# Large-scale survey and database of high affinity ligands for peptide recognition modules

Joan Teyra[1,†] (ID), Abdellali Kelil[1,†] (ID), Shobhit Jain[1,2], Mohamed Helmy[1,‡], Raghav Jajodia[3], Yogesh Hooda[1,#] (ID), Jun Gu[1] (ID), Akshay A D'Cruz[4] (ID), Sandra E Nicholson[4], Jinrong Min[5,6], Marius Sudol[7], Philip M Kim[1,2,8] (ID), Gary D Bader[1,2,8] (ID) & Sachdev S Sidhu[1,8,*] (ID)

## Abstract

Many proteins involved in signal transduction contain peptide recognition modules (PRMs) that recognize short linear motifs (SLiMs) within their interaction partners. Here, we used large-scale peptide-phage display methods to derive optimal ligands for 163 unique PRMs representing 79 distinct structural families. We combined the new data with previous data that we collected for the large SH3, PDZ, and WW domain families to assemble a database containing 7,984 unique peptide ligands for 500 PRMs representing 82 structural families. For 74 PRMs, we acquired enough new data to map the specificity profiles in detail and derived position weight matrices and binding specificity logos based on multiple peptide ligands. These analyses showed that optimal peptide ligands resembled peptides observed in existing structures of PRM-ligand complexes, indicating that a large majority of the phage-derived peptides are likely to target natural peptide-binding sites and could thus act as inhibitors of natural protein–protein interactions. The complete dataset has been assembled in an online database (http://www.prm-db.org) that will enable many structural, functional, and biological studies of PRMs and SLiMs.

**Keywords** domain specificity; peptide inhibitors; peptide library; peptide recognition modules; phage display

**Subject Categories** Methods & Resources; Signal Transduction; Structural Biology

**Mol Syst Biol. (2020) 16: e9310**

## Introduction

Signaling pathways are assembled and regulated by a large network of protein–protein interactions, and currently, more than 409,000 human protein–protein interactions have been identified (Chatr-aryamontri *et al*, 2017). However, the molecular details are not known for most of these interactions.

Most signaling proteins are intrinsically modular and are composed of multiple domains that can fold autonomously (Finn *et al*, 2017). The human proteome is predicted to contain more than 48,000 modular domains that can be grouped into ~6,400 families defined by sequence homology (Punta *et al*, 2012). For ~500 of these families, the protein structure database (PDB) contains structures in complex with peptides, and notably, these peptide recognition module (PRM) families represent ~12,000 of the total predicted modular domains in the proteome (Mosca *et al*, 2014). Structural predictions have also revealed that folded domains are often separated by unstructured regions, which contain many of the short linear motifs (SLiMs) recognized by PRMs (Dinkel *et al*, 2014), and ~100,000 SLiMs in the human proteome have been predicted to interact with PRMs (Tompa *et al*, 2014). Taken together, these large-scale analyses show that while PRMs represent roughly one-tenth of the predicted structural folds, they account for roughly one-third of the predicted domains in the proteome. They are also likely to mediate a substantial fraction of the protein–protein interactions that control human biology. Consequently, a comprehensive analysis of the molecular basis for the function of PRM families would shed light on a substantial proportion of human signaling networks.

PRMs within a family are characterized by structural features that confer a core recognition specificity that is common to most family members (Pawson & Scott, 1997; Kuriyan & Cowburn, 1997). For example, Src-homology-3 (SH3) and WW domains bind proline-

1   The Donnelly Centre, University of Toronto, Toronto, ON, Canada
2   Department of Computer Science, University of Toronto, Toronto, ON, Canada
3   Indian Institute of Engineering Science and Technology, Shibpur, India
4   The Walter and Eliza Hall Institute of Medical Research, Parkville, Vic., Australia
5   Structural Genomics Consortium, University of Toronto, Toronto, ON, Canada
6   Department of Physiology, University of Toronto, Toronto, ON, Canada
7   Department of Medicine, Icahn School of Medicine at Mount Sinai, New York, NY, USA
8   Department of Molecular Genetics, University of Toronto, Toronto, ON, Canada
    *Corresponding author. Tel: +1 416 946 0863; E-mail: sachdev.sidhu@utoronto.ca
    †These authors contributed equally to this work
    ‡Present address: Singapore Institute of Food and Biotechnology Innovation (SIFBI), Agency for Science, Technology and Research (A*STAR), Singapore City, Singapore
    #Present address: MRC Laboratory of Molecular Biology, Cambridge, UK

rich peptides, whereas PSD95/Discs-large/ZO1 (PDZ) domains bind C-terminal sequences (Harris and Lim, 2001). However, sequence and structural differences among members of a PRM family confer peculiarities beyond the core specificity, and these additional features often differ among family members and endow distinct biological functions. Notably, our recent large-scale study of the human SH3 domain family revealed that a significant fraction of these domains exhibit specificities that diverge dramatically from the canonical specificities, suggesting that many SH3 domains may engage in protein interactions that do not rely on polyproline recognition (Kelil *et al*, 2017; Teyra *et al*, 2017). Taken together, these findings emphasize the need for an unbiased and comprehensive approach to the study of PRM specificities for understanding both common and distinct molecular features within families, which in turn give rise to distinct functions for each family member.

Conveniently, both sides of the interactions involving PRMs and SLiMs are relatively simple, making their study highly amenable to a reductionist approach. On the one hand, many PRMs fold autonomously, and they can be purified in a recombinant form independent of the larger proteins in which they naturally reside. On the other hand, large populations of potential SLiMs can be sampled with combinatorial peptide arrays and libraries that can be constructed with well-established combinatorial chemistry and molecular biology methods. Peptide arrays and libraries can be applied *in vitro* to select for SLiMs that bind to isolated PRMs to gain a detailed view of the specificity profile of each PRM independent of its natural protein context. Arrays of synthetic peptides have been used to screen thousands of peptides for binding to PRMs from many families, including PDZ domains (Stiffler *et al*, 2007), SH3 domains (Wu *et al*, 2007; Carducci *et al*, 2012) and SH2 domains (Liu, 2017). These techniques are useful for detailed analysis of known binding specificities and for predicting and validating natural protein partners, but limitations on array size preclude comprehensive exploration of SLiM sequence diversity (Liu *et al*, 2012).

A complementary and much more comprehensive approach is provided by phage display, which enables the sampling of completely unbiased libraries of billions of peptides displayed on phage particles. These large, unbiased libraries permit nearly comprehensive sampling of relatively large stretches of linear peptide sequences and offer a detailed view of optimal binding SLiMs for individual PRMs. Our group and collaborators have improved phage display techniques to increase throughput so that hundreds of PRMs can be analyzed in parallel (Huang & Sidhu, 2011). In the past, we have applied these high-throughput methods for genomescale profiling of three of the largest PRM families; namely, human and worm PDZ domains (Tonikian *et al*, 2008); human, worm and yeast SH3 domains (Tonikian *et al*, 2009; Xin *et al*, 2013; Teyra *et al*, 2017); and human and worm WW domains (unpublished). These studies have provided 7,063 unique and validated peptide binders for a total of 342 domains, and the resulting specificity maps have revealed the versatile and specific nature of these modules and their interactions. Moreover, the database of optimal ligands for PDZ, SH3 and WW domains has proven to be highly useful for predicting and validating natural interactions, developing intracellular inhibitors of natural interactions, and guiding structural studies to better understand the details of molecular recognition (Schon *et al*, 2002; Slivka *et al*, 2009; Zhang *et al*, 2009; Kelil *et al*, 2016; Hershey *et al*, 2016).

Here, we report a broad survey of specificity profiles for diverse PRM structural families, using the phage-displayed peptide library technology that we previously applied to the PDZ, SH3 and WW families. In total, we report 1,091 unique peptides, each binding specifically to one of 163 unique PRMs representing 79 distinct structural families. We have compiled the new data with previous phage-derived data for PDZ, SH3 and WW domains to assemble an online database containing 7,984 unique, validated peptide ligands for 500 PRMs representing 82 structural families. The online database and search tools (http://www.prm-db.org) enable the research community to scan proteomes for putative protein interactors, identify the structural interactions between PRMs and SLiMs, and access potent peptidic inhibitors of hundreds of PRMs. Thus, the compiled and searchable database should facilitate many structural, functional and biological studies of PRMs and SLiMs. We also present structural and functional analyses that demonstrate the utility of our PRM-peptide database, paving the way for future studies that will lead to a deeper understanding of proteins and biological systems containing PRMs and SLiMs.

# Results

### A panel of diverse PRM structural families

To assemble a panel of recombinant protein domains representing diverse PRM structural families, we conducted a proteome-wide computational survey of protein folds to identify those that have been confirmed to bind peptides based on structural evidence. We used the Pfam database (Punta *et al*, 2012) to retrieve all domains from human proteins in the UniProt database (The UniProt Consortium, 2012) based on high-quality Pfam-A domain definitions. This process yielded 48,091 domains that were grouped in to 6,483 unique families based on sequence conservation (Dataset EV1). Within this large set, we defined PRM families as those that contained at least one member for which a structure in complex with a peptide was deposited in the Protein Data Bank, and this yielded 12,784 domains (Dataset EV1) grouped into 503 structural families (Dataset EV2). For our panel, we chose 246 domains from 90 of these confirmed PRM families (Dataset EV3). In addition to confirmed PRM families, we also wanted to explore putative PRM families for which structural evidence of peptide binding does not yet exist, but which may bind peptides due to structural similarity to confirmed PRM families. Thus, we added to our panel 39 domains from 25 families (Dataset EV3), each of which was grouped in a clan with a confirmed PRM family, with a clan being defined by Pfam as a group of families with an evolutionary relationship based on structure, function and sequence comparison (Finn *et al*, 2016). We prioritized cases where there was some biological evidence in the literature that the family could bind to SLiMs in proteins. In general, we selected more domains from larger families and included all members of three families: hormone receptor, DEP and GYF.

In total, our panel contained 285 human protein domains representing 115 different structural families, including 90 confirmed and 25 putative PRM families (Dataset EV3). In order to generate encoding DNA sequences, we defined domain boundaries by alignment with the available structure sharing the most sequence similarity

with each domain, and we optimized the nucleotide sequence for expression in *Escherichia coli*. Each gene fragment was fused to the end of a gene encoding glutathione S-transferase (GST), and the resulting open-reading frames were expressed in *E. coli* to produce GST-PRM fusion proteins. Using standardized and previously described high-throughput methods (Huang & Sidhu, 2011), we successfully purified 215 of the 285 GST-domain fusion proteins, as evidenced by major bands migrating at the predicted molecular weight visualized by SDS–PAGE and quantified by Bradford assay (Dataset EV3).

## A catalog of PRM binding specificities

To conduct the study of PRM binding specificities, we constructed a highly diverse phage-displayed library of $4 \times 10^{10}$ random hexadecapeptides. The library was encoded by a degenerate mutagenic oligonucleotide synthesized with special trinucleotides that allowed for an approximately equal representation of 19 genetically encoded amino acids but excluded cysteine to avoid complications due to potential disulfide bonds. The library was cycled through five rounds of high-throughput selections with the panel of 215 purified proteins, as described previously (Huang & Sidhu, 2011). For most of the domains, we analyzed 48 individual peptide-phage clones from rounds 4 and 5 with phage ELISAs, and the positive clones that bound to target protein but not to GST were sequenced. We obtained one or more unique binding peptides for 163 domains representing 79 distinct structural families (Dataset EV3) and compiled a dataset containing a total of 1,091 unique peptide ligands. We have deposited the expression plasmids for most of the PRMs in the Addgene plasmid repository, and all associated peptides are listed in Dataset EV4 with their ELISA values.

In order to analyze and visualize peptide-binding specificity profiles for each PRM, we generated a clustering and gap-free sequence alignment of the set of peptides (see Methods), since the vast majority of binding peptide motifs are known to be linear in sequence (Stein & Aloy, 2010). The peptides that clustered as outliers were not considered in the alignment under the assumption that they were binding in a different mode or to a different region than the rest, as previously observed for other PRMs (Teyra *et al*, 2017). Each alignment was used to create a position weight matrix (PWM), in which the weight of each amino acid at each position of the matrix equaled its frequency multiplied by the difference between the frequencies of the most and least abundant amino acids. Thus, each column in the matrix depicted the amino acid binding preference of the PRM at the ligand position as a weighted frequency distribution (see Methods). For each PWM, the specificity profile was visually represented as a sequence logo (Schneider & Stephens, 1990), where the relative sizes of the letters indicate their weighted frequencies, and the total height of the letters indicates the specificity of the PRM for that position.

The specificity at each logo position was quantified using the specificity potential (*SP*) score, which is defined as the sum of the amino acid weighted frequencies, and ranges from least specific (any amino acid is recognized, $SP = 0$) to most specific (a single amino acid is recognized, $SP = 1$) (Tonikian *et al*, 2008). In the phage-displayed peptide library, randomized hexadecapeptides are flanked by invariable glycine residues that can also participate in PRM recognition. Thus, in cases where logos had specific positions

located at the extreme N- or C-terminal position, a glycine residue was added to the aligned peptide sequences at that terminus, and new PWMs and logos were generated. To select the core significant portion of the logo, we trimmed flanking positions with low *SP* values (< 0.2), and the final specificity profile was used to calculate the total specificity potential score ($SP^t$), which was defined as the sum of all *SP* scores across the remaining positions in the logo (Tonikian *et al*, 2008). For 74 PRMs representing 44 structural families, we had enough peptide data ($n \geq 5$) to calculate reliable specificity scores ($SP^t > 1$) and generate high confidence specificity profiles (Fig 1).

Taken together, we were successful in purifying 75% (215 of 285) of domains attempted, and for these, we were successful in generating binding peptides for 76% (163 of 215). These success rates were similar to those achieved in previous large-scale specificity profiling studies of PDZ (Tonikian *et al*, 2008), SH3 (Teyra *et al*, 2017) and WW domains (unpublished). In total, we were able to isolate 1,091 unique peptides for the 163 PRMs, and we had enough peptide data to generate specificity profiles for 74 PRMs, which we used for further statistical analysis. Failure to purify 70 of the 285 domains may be due to non-optimal boundaries for the expression constructs or instability of the domains in isolation from the full-length protein. Failure to identify binding peptides for 52 of the 215 purified domains may be due to weak affinities that cannot be selected by the phage display method, or specificity for post-translational modifications (PTMs) that cannot be mimicked by standard amino acids.

## General features of PRM specificity profiles

The discovery of new peptides and motifs mediating interactions has been of intense interest for many decades (Dinkel *et al*, 2014). Despite many efforts, our current knowledge covers only a small fraction of the thousands of motifs that are predicted to exist in the human proteome (Tompa *et al*, 2014; Kelil *et al*, 2016), mostly because of the slow and arduous nature of low-throughput motif identification and characterization (Gibson *et al*, 2015). Alternative large-scale approaches, such as peptide arrays and phage display, have been of great utility to identify new peptide partners and binding specificity profiles, but until now, studies have focused on members of single families (Teyra *et al*, 2012). For the first time, our rich source of peptide information on a diverse set of 74 PRMs covering 44 structural families allowed us to obtain a broad overview of the factors dictating PRM specificities and to compare these general features to those observed in the previously characterized PDZ, SH3 and WW domain families.

Toward this aim, we analyzed several features for our 74 PRM specificity profiles, including length, specificity, amino acid composition and hydrophobicity (Fig 2). For comparison, each position in each profile was classified empirically as either specific ($SP > 0.33$) or non-specific ($SP < 0.33$) (Appendix Fig S1). For comparison with the previously studied PRM families, we used binding peptide sequences for 58 human PDZ domains (Tonikian *et al*, 2008), 115 human SH3 domains (Teyra *et al*, 2017), and 66 human WW domains (unpublished) to calculate PWMs and logos with the approach described above (Appendix Fig S2). Also, because disordered regions in proteins are believed to contain most of the natural peptide ligands that bind to PRMs (Tompa *et al*, 2014), we used

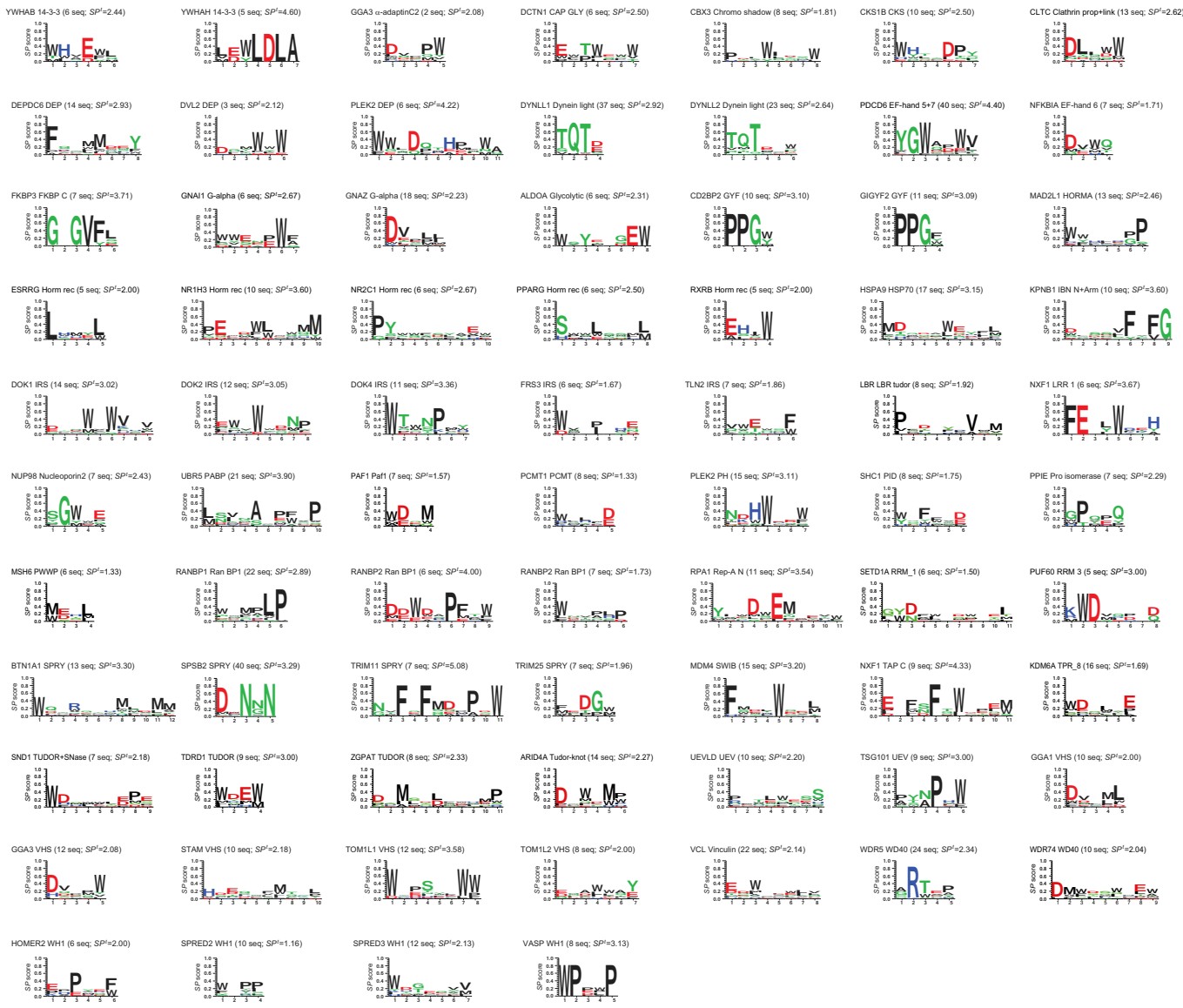

**Figure 1. Specificity profiles for PRMs.**

The specificity profiles for 74 PRMs are represented as logos showing the preferences at each peptide position. The following information is provided above each logo: name of the protein from which the PRM was derived, PRM family name, number of peptide sequences used to derive the logo (seq), and total specificity potential score ($SP^t$) for the logo. Logos are presented in alphabetical order by family.

IUPred2A (Mészáros *et al*, 2018) to computationally identify 26,283 disordered regions within the human proteome, and we defined this as a natural "disorderome" set for comparison with our phage-derived database of optimal peptide ligands for PRMs.

For each PRM, we defined the profile length as the total number of positions in the logo and quantified the profile specificity by determining the $SP^t$ score across each position. Our analysis showed that the members of the PDZ, WW, and SH3 domain families exhibit a wide range of motif lengths (Fig 2A) and specificities (Fig 2B) but with some distinct features. The typical motifs for SH3 and WW domains are known to have three highly conserved positions, but canonical SH3 motifs (RxxPxxP or PxxPxR) are longer than canonical WW motifs ([LP]PxY) (Macias *et al*, 2002) and often also show preferences for Leu residues at additional positions (Teyra *et al*, 2017). Moreover, many SH3 domains exhibit non-canonical specificities with motif lengths longer than those of canonical motifs (Teyra *et al*, 2017). Thus, on average, SH3 domains exhibited longer motifs and higher specificities than WW domains. PDZ domains typically recognize C-terminal ligands but can interact with many residues preceding the last residue, and consequently, the analysis showed diverse motif lengths for the various family members, including some that are very long and very specific (Tonikian *et al*, 2008). Given the wide diversity of structural folds and consequent specificity profiles for the new set of 74 PRMs, it is not surprising

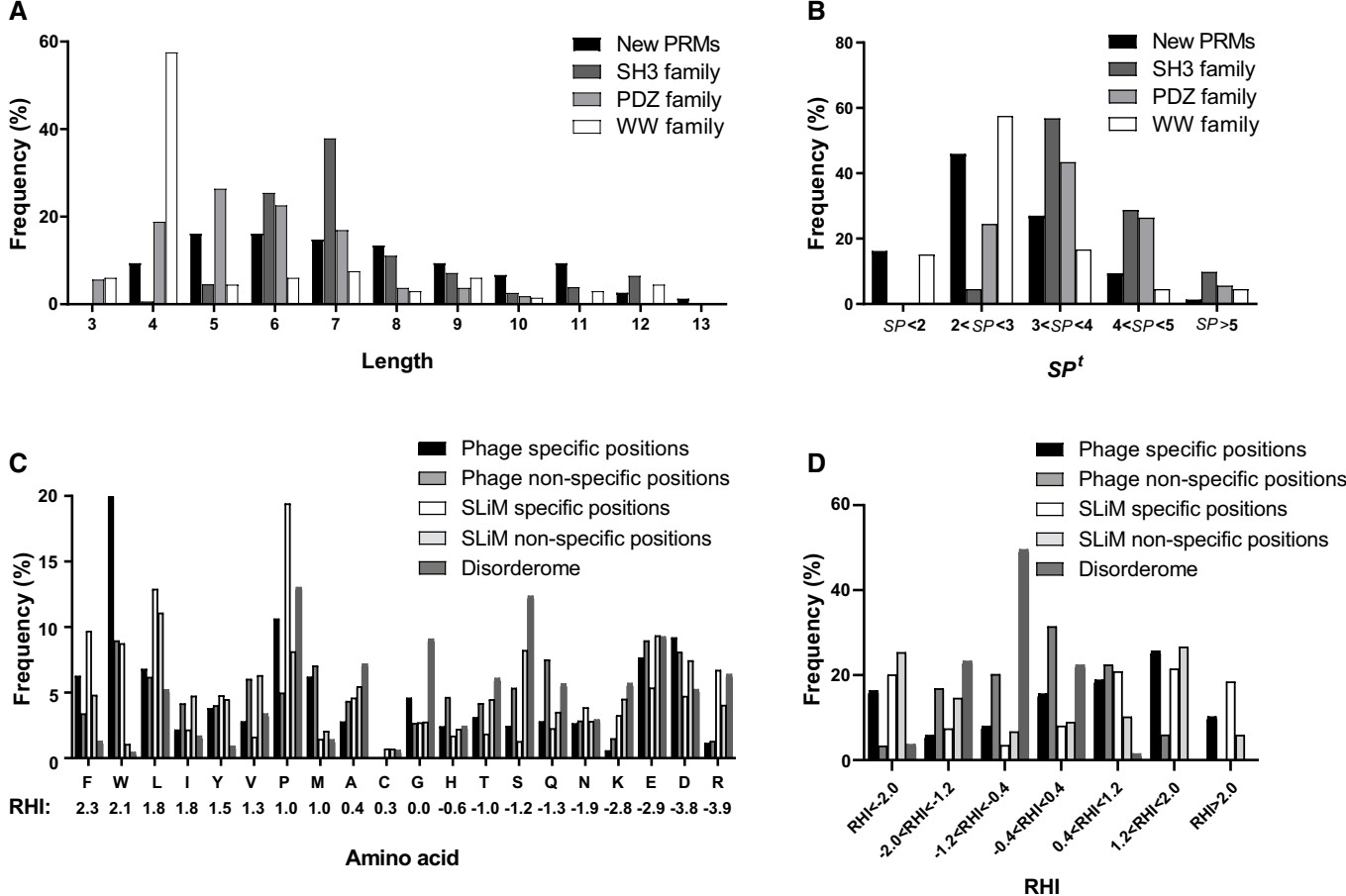

**Figure 2.  Physicochemical properties of PRM ligands.**

Data are shown for the 74 PRMs for which specificity profiles were determined in this study (new PRMs, Fig 1), for human SH3, PDZ and WW domains studied previously (Tonikian *et al*, 2008; Teyra *et al*, 2017), and for SLiMs and disorderome peptides.

A   Distribution of specificity profile lengths.
B   Distribution of total specificity potential ($SP^t$) scores.
C   Frequencies of amino acids at specific and non-specific positions in phage-derived peptides and SLiMs, and in disorderome peptides. Amino acids are ordered from highest to lowest hydrophobicity measured by Roseman's Hydropathy Index (RHI), which is shown below each amino acid denoted by the single-letter code.
D   Distribution of the RHI for specific and non-specific positions in phage-derived peptides and SLiMs and in disorderome peptides.

that these exhibit a diversity of motif lengths with no distinct length dominating (Fig 2A). On average, the new PRMs appear to be less specific than SH3 and PDZ domains and are comparable to WW domains (Fig 2B).

To assess the relative contributions of the different amino acids in peptide ligands for recognition by PRMs, we plotted the frequencies of the amino acids in a total of 277 specific and 267 non-specific positions within the 74 PRM specificity profiles (Fig 2C and Appendix Fig S1). We also compared the characteristics of the phage-derived peptides with the disorderome and with human SLiMs from the eukaryotic linear motif (ELM) repository (Kumar *et al*, 2020), which contains manually curated information for experimentally validated natural SLiMs and reflects natural PRM binding preferences. For better comparison with our results, we differentiated between specific and non-specific positions in the SLiMs, defined as positions that show preferences for a subset of residues and those that can tolerate any residue, respectively.

By far, Trp was the most frequent amino acid at specific positions in our data (20%) (Fig 2C), which agrees with what has been shown for human SLiMs (Davey *et al*, 2012), for which our calculations show an abundance of 8.8%. Differences might be attributed to the fact that the ELM repository does not contain SLiM instances for most of the families that recognize Trp-containing peptides (e.g., CAP-GLY, Chromo Shadow, Glycolytic, PH, SWIB and VHS families, Fig 1). This high frequency is even more striking, considering the very low abundance of Trp in the disorderome (0.4%) (Fig 2C). Aromatic Trp side chains are often buried at interfaces, where the indole ring can form stacking interactions with other aromatic residues and cation-π interactions with Arg side chains, and the nitrogen group in the indole ring can form hydrogen bonds with polar residues (Betts & Russell, 2003). Hydrophobic Leu (6.8%), Phe (6.3%) and Met (6.2%) residues were also highly prevalent at specific positions, whereas Phe and Met residues were relatively rare in the disorderome. A similar profile for hydrophobic residues was

observed for specific positions in SLiMs, although the prevalence of Leu and Phe was 1.9- or 1.5-fold higher, respectively, compared with phage-derived ligands. The second most frequent amino acid at specific positions was Pro (11%), and Pro residues were also most abundant in specific positions of SLiMs (19%) and were highly prevalent in the disorderome (13%) (Fig 2C). Notably, the frequency of proline at specific positions would be even greater if we had included the large families of polyproline-binding SH3 and WW domains in the analysis. In addition to providing hydrophobic contact surface through its side chain, the rigidity of the proline main chain may minimize the loss of conformational entropy upon binding (Kay *et al*, 2000). Although hydrophilic residues were less abundant than hydrophobic residues at specific positions, Asp (9.2%) and Glu (7.7%) residues were highly prevalent and were much more abundant than Arg (1.2%) and Lys (0.6%) residues, indicating that PRMs generally make more specific contacts with negatively charged rather than positively charged ligand side chains. SLiMs also showed lower abundance of hydrophilic residues compared with hydrophobic residues, but charged residues were fairly abundant at specific positions. However, unlike for phage-derived peptides, the specific positions of SLiMs showed no significant differences between the abundances of negatively charged Asp (4.7%) and Glu (5.4%) residues compared with positively charged Arg (6.7%) and Lys (3.3%) residues (Fig 2C). Together, the seven most abundant amino acids (Trp, Pro, Asp, Glu, Leu, Phe, Met) account for more than two-thirds (67.2%) of the amino acids at specific positions of phage-derived peptides, and this bias indicates that peptide ligands often rely on conformational rigidity conferred by Pro residues, and interact with PRMs mainly through hydrophobic interactions mediated by Trp, Phe, Leu, Met and Pro residues, but also rely on electrostatic interactions and hydrogen bonds mediated by Asp and Glu residues.

We used Roseman's Hydropathy Index (RHI) (Roseman, 1988) to assess the hydrophobicities of the amino acid frequencies for the 277 specific and 267 non-specific positions of the 74 phage-derived PRM specificity profiles and for the 1,633 specific and 7,772 non-specific positions of 1,385 SLiMs. We also calculated the mean RHI for each of 160,713 16-mer peptides derived from the human disorderome by determining the average of the RHI values for the 16 residues in each peptide (Fig 2D; Roseman, 1988). Gly has a benchmark RHI of zero, as it lacks a side chain, and amino acids with values above or below zero are considered hydrophobic or hydrophilic, respectively (Fig 2C). Binning based on the RHI values (Fig 2D) revealed that most disorderome peptides are more hydrophilic than glycine (mean RHI = −0.84), whereas in contrast, most specific positions in the phage-derived ligands are much more hydrophobic (mean RHI = 0.21) and even non-specific positions are more hydrophobic (mean RHI = −0.23) than the disorderome, although RHI distribution for specific positions is wider than for non-specific positions. Taken together with the results of the amino acid distributions, these results show that optimal peptide ligands for PRMs are characterized by much greater hydrophobicity than natural peptides in the disorderome, and the hydrophobicity is highest in specific positions that presumably make contacts with PRM surfaces. Much of the increased hydrophobicity is attributable to highly hydrophobic Trp and Phe residues, which are abundant at specific positions of peptide ligands but are rare in the disorderome. Moreover, our analysis showed that, overall, SLiMs are much more

hydrophilic than phage-derived peptides (mean RHI = −0.65 and 0.03, respectively) and slightly less hydrophilic than the disorderome (mean RHI = −0.84). However, specific positions of SLiMs are more hydrophobic than those of phage-derived ligands (mean RHI = 0.78 and 0.21, respectively), due to high prevalence of Phe, Leu and Pro in SLiMs. In contrast, non-specific positions of SLiMs are much more hydrophilic than those of phage-derived ligands (mean RHI = −2.1 and −0.23, respectively). Overall, phage-derived peptides reflect the hydrophobic character of SLiMs at specific positions, which are critical for PRM recognition.

## Structural rationalization of PRM-ligand interactions

To gain insights into the molecular basis for the interactions between PRMs and their cognate phage-derived peptide ligands, we took advantage of the extensive Protein Data Bank (PDB, www.rcsb.org) repository of structures of PRMs in complex with peptide ligands. For each of 163 query PRMs for which we obtained phage-derived peptide ligands, we searched the PDB for homologous proteins in complex with ligands containing 5–30 amino acids, presuming that these represented structures of PRMs in complex with peptides. We calculated the percent sequence identity between the query PRM and each structure PRM. We also calculated the percent similarity between the structure peptide and the most similar phage-derived peptide in our database, using amino acid groups clustered on the basis of similar physicochemical properties (see Methods), since amino acids with common physicochemical properties can be exchangeable under particular structural environments (Taylor, 1986). For each PRM/peptide pair in our database, we selected the PRM/peptide pair in the PDB that exhibited the highest sequence similarity between the peptides and high sequence identity between the PRMs. We examined each structure and identified the PRM/peptide residues that formed the interface (side chain atoms ≤ 5 Å away from each other) and also calculated the percent sequence identity and similarity between these residues and the corresponding residues for the analogous PRM and peptide in our data, respectively, since these binding-site residues play key roles for determining the specificity and affinity of peptide recognition (Tonikian *et al*, 2008; Liu *et al*, 2010; Stein & Aloy, 2010; Gorelik *et al*, 2011).

We were able to identify matches in the PDB for 135 of the 163 PRMs (63 with specificity logos and 72 without, Dataset EV5), with 45 and 63% having a near-perfect match (> 98% sequence identity) when considering the full sequence or the binding-site sequence, respectively (Fig 3A). However, we could not identify a representative PDB structure with > 10% sequence identity for 28 of the 163 PRMs in our database (Dataset EV3), and a structural understanding of these PRM/peptide interactions will require the elucidation of new structures. For each of the 135 PRMs with a matched PRM-ligand structure in the PDB, we compared the structure peptide to the most similar phage-derived peptide with the assumption that similar sequences are likely to use similar molecular interactions to bind to similar sites on PRMs. For the 63 PRMs with specificity logos derived from multiple phage-derived peptides (Fig 1), we considered only the regions of the peptides that aligned with the logo. For the 72 PRMs without specificity logos, we considered the length of structure peptide spanning those residues that were in contact with the PRM. Analysis of the 135 matched PRMs revealed that 88% of the phage-derived peptides (119 of 135) exhibited > 40% similarity

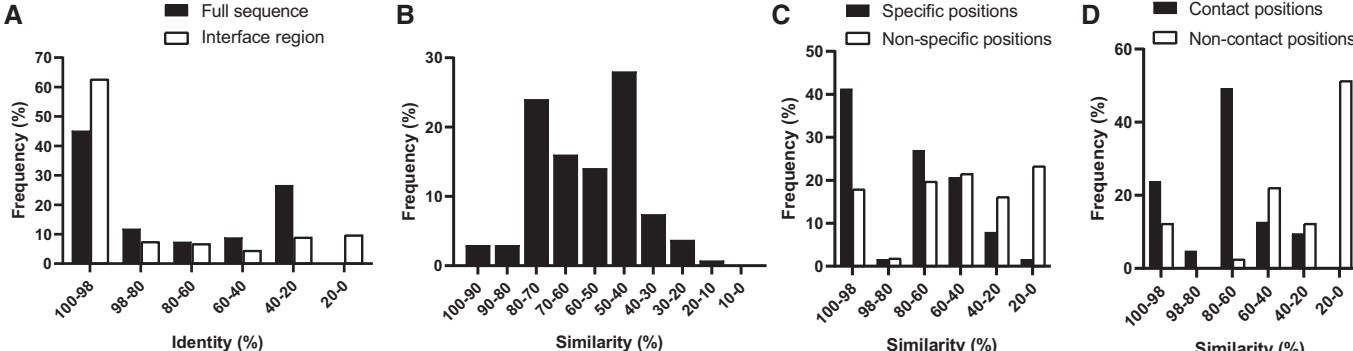

**Figure 3. Comparison of PRMs and phage-derived ligands with matched PRM-ligand complex structures.**

Data are shown for 135 of 163 PRMs that yielded at least one phage-derived ligand and for which the PDB contained at least one PRM with > 10% sequence identity, and these are compared with the best-matched PRM/ligand complex in the PDB (Dataset EV5).

A  Distribution of sequence identities between the studied PRMs and the matched PRM structures for the full sequence or the sequence of the peptide-binding interface region.
B  Distribution of sequence similarities between the ligand in the complex structure and the most similar phage-derived peptide for all 135 PRMs.
C  Distribution of sequence similarities between the ligand in the complex structure and the most similar phage-derived peptide for the 63 PRMs with specificity profiles (Fig 1), either at specific positions or non-specific positions.
D  Distribution of sequence similarities between the ligand in the complex structure and the most similar phage-derived peptide at contact and non-contact positions in the structure.

with their corresponding structure peptides (Fig 3B). Considering only the 63 PRMs with specificity logos, the phage-derived and structure peptides exhibited an average similarity of 75 and 50% at specific and non-specific positions in the profile logo, respectively (Fig 3C), suggesting that specific positions likely represent residues that contribute favorably to the interaction with the PRM, and are thus conserved between the phage-derived and structure peptides. Moreover, the phage-derived and structure peptides exhibited much higher average sequence similarities at positions that contact the PRM in the structure (75%), compared with non-contact positions (29%), which highlights the functional importance of positions that are similar between peptide pairs (Fig 3D).

Taken together, these results suggest that the phage-derived peptides for most of the PRMs in our database likely represent ligands that bind to functional sites identified previously in related PRM structures in the PDB. Differences between optimal phage-derived peptides and natural structure peptides may also be of interest to understand the types of substitutions that can enhance the affinities of natural ligands and could thus be useful for inhibitor design. Consequently, our phage-derived peptides can provide molecular insights into natural protein function and can be used as inhibitors of natural protein–protein interactions.

## Phage-derived mimics of peptides containing PTMs

To visualize each PRM/peptide interaction at the structural level, we depicted the structure of each matched PRM-ligand from the PDB, along with the sequences of the structure peptide and the most similar phage-derived peptide and the specificity logo, if available. For 118 of the 135 phage-derived peptides, we could identify similar structure peptides that did not contain any PTM, and these represented 55 PRMs with enough phage-derived ligands to derive specificity logos (Fig 4) and 63 PRMs without specificity logos (Appendix Fig S3 and Dataset EV5). Notably, the remaining 17 phage-derived peptides could only be matched with structure peptides that contained a PTM, and these represented 8 or 9 PRMs with or without specificity logos, respectively (Fig 5).

The 17 structure peptides with PTMs were divided into three groups of eight, four or five peptides containing phosphorylated serine/threonine (pSer/pThr, Fig 5A), phosphorylated tyrosine (pTyr, Fig 5B) or methylated Arg/Lys (meArg/meLys, Fig 5C), respectively. Six of the eight PRMs that recognized pSer/pThr belonged to the 14-3-3 domain family and the other two belonged to the CKS or NIF domain family. In five of these, the aligned phage-derived peptide contained a negatively charged Asp/Glu

**Figure 4. Comparison of phage-derived ligands and structures of peptides in complex with PRMs.**

Depicted are the 55 PRMs for which phage-derived specificity profiles were determined (Fig 1) and for which the structure peptide did not contain a PTM. The name of the protein from which the studied PRM was derived is listed at the top with the PRM family name in parenthesis, followed by the specificity profile logo determined from phage-derived peptides, and sorted alphabetically by family. The alignment below the logo shows the sequences of a phage-derived peptide (top) and the peptide ligand from the best-matched PRM-ligand complex structure in the PDB (bottom). Similar residues in the two peptides are shaded gray and residues that make contact with the PRM in the structure are underlined. The structure of the best-matched PRM-ligand complex in the PDB is shown with the PRM and peptide ligand main chains rendered as gray or green ribbons, respectively. Red and blue spheres denote ligand positions that are similar to the phage-derived peptide and are contact or non-contact positions, respectively. The peptide main chain is only depicted for those residues that are shown in the alignment with the phage-derived peptide. The PDB entry code is shown above each structure.

▶

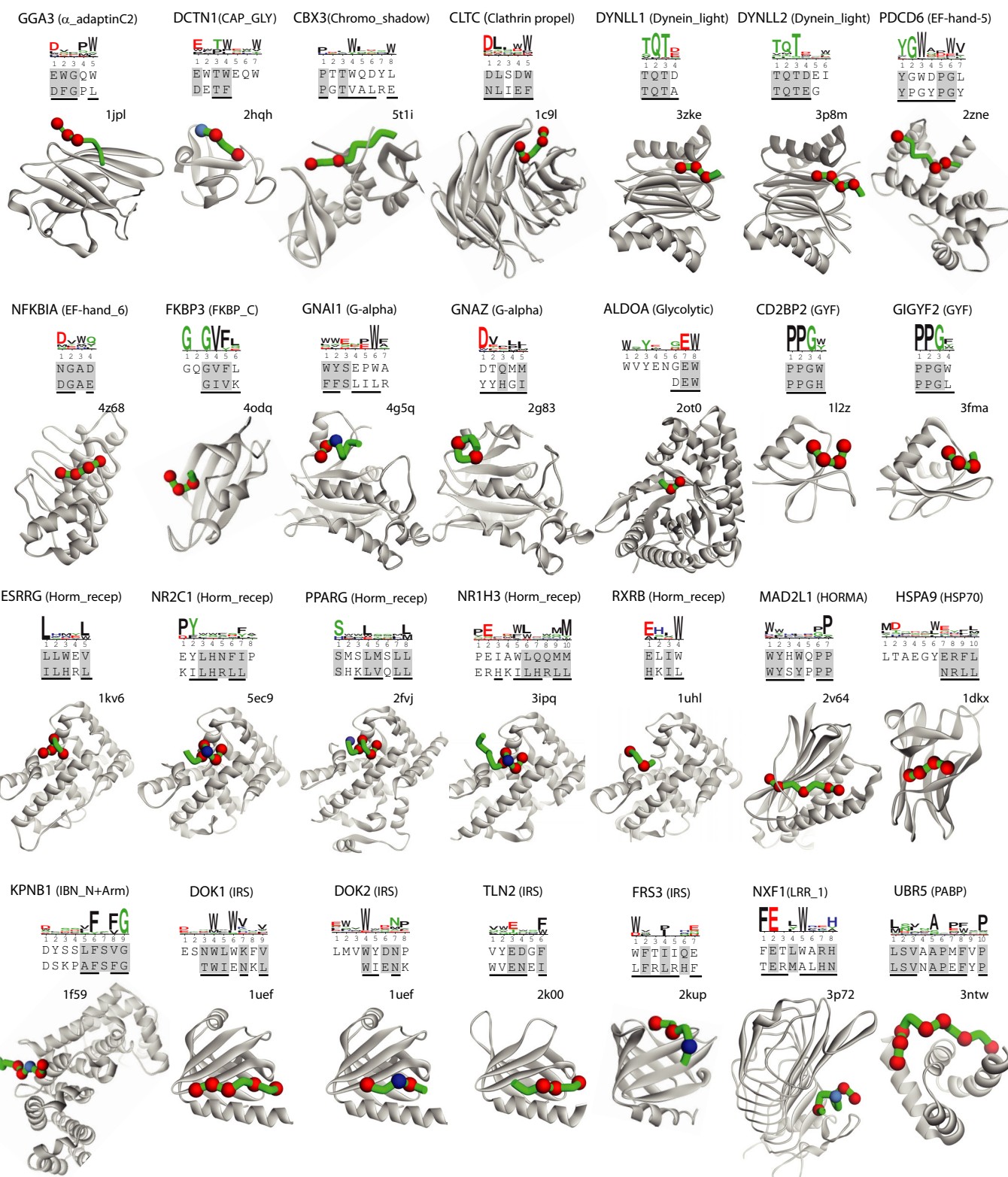

**Figure 4.**

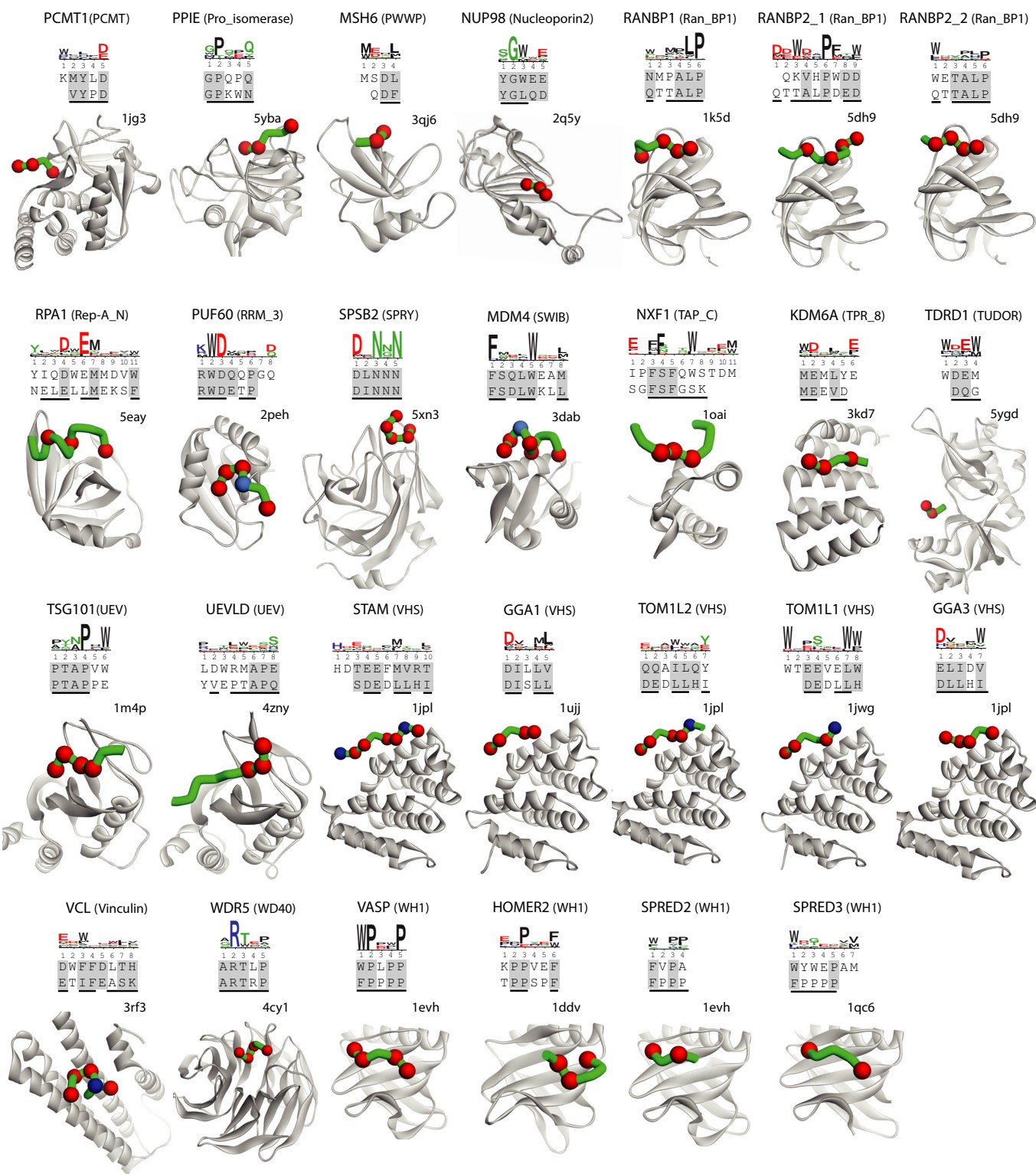

**Figure 4.**

residue in place of the pSer/pThr residue in the structure peptide, consistent with other reports that have shown that Asp/Glu can effectively mimic the shape and charge of pSer/pThr (Sundell *et al*, 2018). For two of the other PRMs, the 14-3-3 domains of YWHAE and YWHAZ, the phage-derived peptide contained an aromatic Tyr/Trp residue in place of pSer/pThr. The structure peptide for the remaining PRM, the NIF domain of CTDSP2, was unusual in that it contained two pSer residues and exhibited only minimal

homology with the phage-derived peptide, thus making it unclear whether the phage-derived peptide bound to the same site as the structure peptide. Three of the four PRMs that recognized pTyr belonged to the IRS domain family and the fourth belonged to the PID family, and in each case, the alignment showed that the phage-derived peptide contained a hydrophobic residue in place of the pTyr in the structure peptide. Finally, the five PRMs that recognized meArg/meLys included two TUDOR domains, a PHD domain, a TUDOR-knot domain and a WD40 domain. Except for the WD40 domain, the alignments revealed that each phage-derived peptide substituted a hydrophobic residue for the meArg/meLys residue in the structure peptide. In the case of the WD40 domain, meLys in the structure peptide was substituted by His in the phage-derived peptide, but in this case, the structure peptide showed low similarity with the phage-derived peptide and specificity logo, making it uncertain whether the two peptides recognize the same site in the same manner. Taken together, these results showed that phage-derived peptides without PTMs can mimic peptide ligands that contain PTMs in many cases, either by using Asp/Glu residues that mimic pSer/pThr residues or by using hydrophobic residues that likely act as partial mimics of PTMs. Thus, our results could be useful for designing PTM mimics, but further biophysical and structural studies will be necessary to reveal the molecular basis for PTM mimicry.

### Comparison of phage-derived ligands with natural ligands

To assess the biological relevance of the phage-derived ligands, we compared our database to the ELM repository (Kumar *et al*, 2020), which contains manually curated information for experimentally validated natural SLiMs with their corresponding PRM interacting partners. The SLiMs are grouped into ELM classes or motifs described by regular expressions that capture the key features of SLiMs matching the sequence patterns, and one or several classes may represent the ligand-binding specificities for each PRM structural family. Since the number of SLiMs used to generate ELM motifs is very limited, some ELM motifs represent only a single SLiM, and many motifs might be too specific to capture the broad binding preferences of all the members of a PRM family. Therefore, we carried out a comparative analysis for only those PRMs with annotated SLiM ligands in the ELM repository, since the ELM motifs generated for these PRMs would capture their natural SLiM ligands, and consequently, they would be the most reliable motifs to compare with our phage-derived results.

Of the 500 PRMs in our database and the 163 PRMs in this study, we found that only 44 or 20, respectively, had at least one SLiM ligand, showing low coverage of our PRMs in the ELM repository (Dataset EV6). In order to compare the highest resolution examples of ELM and phage-derived results, we focused our analysis on PRMs for which our database contained enough phage-derived peptides to generate specificity profiles and for which the ELM class associated with the SLIM ligands did not contain any PTMs. For these eight out of 20 PRMs, we compared the ELM motif with the most similar phage-derived peptide and with the specificity profile, and we rationalized the comparisons using PRM structures with bound peptides (Fig 6).

For six PRMs—TSG101(UEV), SPSB2(SPRY), VASP(WH1), UBR5 (PABP), WDR5(WD40) and CD2BP2(GYF)—the ELM motifs agreed closely with their respective phage-derived specificity profiles and

with the structures of PRM-ligand complexes (Fig 6). The TSG101 (UEV) structure contains a groove that recognizes the Pro-Thr-Ala-Pro sequence of the ligand, and $Pro^4$ is the most buried residue, and also, the most conserved sequence in the specificity profile. The SPSB2(SPRY) structure contains a hydrophilic pocket that recognizes a peptide loop representing the ELM motif. Notably, positions 2 and 4 are the most solvent accessible in the structure and also the least conserved in the specificity profile, but nonetheless, several phage-derived peptides matched the ELM motif exactly. In the VASP (WH1) structure, three hydrophobic pockets are occupied by $Phe^1$, $Pro^2$ and $Pro^5$ residues in the peptide ligand, and these residues are very similar to the ELM motif and the phage-derived specificity profile, which contain $Trp^1$, $Pro^2$ and $Pro^5$. The Pro residues at positions 3 and 4 of the structure peptide are exposed to solvent, consistent with lower conservation of these positions in the phage-derived specificity profile. The UBR5(PABP) structure reveals an extended binding site that makes contacts with six residues imbedded within a 10-residue stretch of the peptide ligand. The importance of these six contact positions is reflected in the ELM motif, and also in the phage-derived peptides and specificity profile, which all show good agreement. The WDR5(WD40) structure contains a deep cavity that accommodates an Arg residue at position 2, which is completely conserved in the ELM motif and the phage-derived specificity profile. The ELM motif extends across seven positions and it closely matches the structure peptide and the phage-derived specificity profile. Finally, the CD2BP2(GYF) structure contains a hydrophobic cleft that interacts with a central Pro-Pro-Gly sequence in the peptide ligand as well as flanking residues on both ends. The short phage-derived specificity profile shows strong conservation for a Pro-Pro-Gly sequence followed by an aromatic residue, and notably, a previous study with phage display and peptide arrays defined a very similar tetrapeptide specificity profile that was validated by proteomic experiments (Kofler *et al*, 2005). Consequently, the short phage-derived specificity profile closely matches the core of the structure peptide and ELM motif, suggesting that this central region is most important for binding. Taken together, the comparisons of the eight PRMs with SLiM ligands in the ELM repository showed that our phage-derived peptides and specificity profiles closely match the ELM motifs in six cases, confirming that most phage-derived peptides resemble natural SLiMs, and thus, likely bind to known functional sites on their cognate PRMs.

### Online database of phage-derived PRM specificity profiles

We have developed an online database (http://www.prm-db.org) that provides access to the new data reported here, and also, to previous phage-derived data for PDZ, SH3 and WW domains (Tonikian *et al*, 2008; Xin *et al*, 2013; Teyra *et al*, 2017). In total, the online database contains 7,984 unique peptide ligands, each validated to bind one of 500 PRMs representing 82 structural families. The database can be queried by protein or family name and the search can be restricted by species. For each PRM, the database provides the specificity logo, the top three phage-derived peptide ligands, and the best-matched PRM-ligand structure in the PDB. Users can also download the full list of peptide ligands, the PWM matrices containing the amino acid frequencies, and a list of the best-matched PRM-ligand structures in the PDB. Moreover, the database provides a list of human proteins matching PRM binding

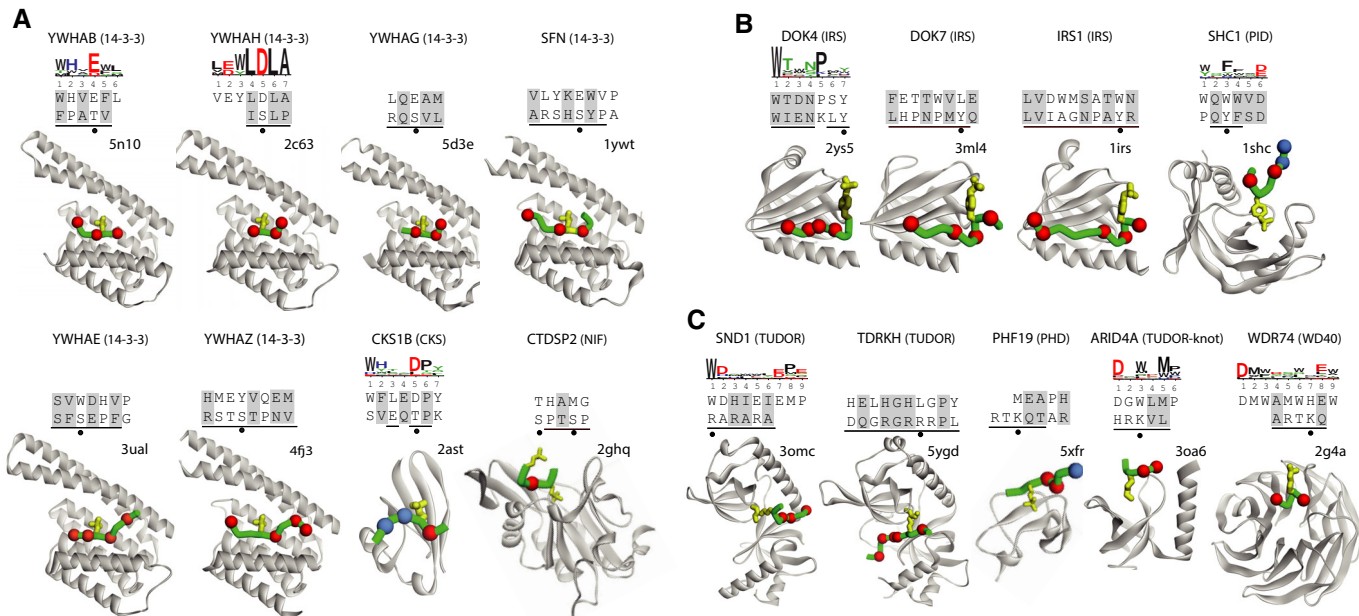

**Figure 5.** Comparison of phage-derived ligands and structures of PTM-containing peptides in complex with PRMs.

A–C Depicted are the 17 PRMs for which the ligand in the structure contains (A) pSer/pThr, (B) pTyr, or (C) meArg/meLys. The name of the protein from which the studied PRM was derived is listed at the top with the PRM family name in parenthesis, followed by the specificity profile determined from phage-derived peptides, if available. Below, the alignment shows the sequences of a phage-derived peptide (top) and the peptide ligand from the best-matched PRM-ligand complex in the PDB (bottom). Similar residues in the two peptides are shaded gray and residues that make contact with the PRM in the structure are underlined. Filled circles below the alignment indicate residues that contain PTMs. The structure of the best-matched PRM-ligand complex from the PDB is shown with the PRM and peptide ligand main chains rendered as gray or green ribbons, respectively. Red and blue spheres denote ligand positions that are similar to the phage-derived peptide and are contact or non-contact positions, respectively. Side chains that contain PTMs are shown as yellow sticks. The peptide main chain is only depicted for those residues that are shown in the alignment with the phage-derived peptide. The PDB entry code is shown above each structure.

motifs, which was generated by scanning the human proteome with the PWMs. Combined with other experimental data, these can help to prioritize biological experiments to explore putative natural interactions (Jain & Bader, 2016). Thus, the compiled and searchable database should facilitate many structural, functional and biological studies of PRMs and SLiMs.

## Discussion

We have compiled a broad and large-scale specificity map for PRMs, encompassing 1,091 unique peptide ligands and 163 unique PRMs representing approximately one-sixth of the 503 predicted PRM families encoded by the human genome. Analysis of 74 PRMs with specificity profiles revealed that binding specificities often differ substantially across structural families. Analysis of amino acid composition and hydrophobicity showed that optimal peptide ligands for PRMs are overall more hydrophobic than disordered regions of human proteins, and hydrophobicity is highest in specific positions that presumably interact directly with PRMs. This hydrophobic bias of the phage-derived peptides has been reported previously in a study of PDZ domain specificity profiles (Luck & Travé, 2011), in which they observed that phage-derived specificity profiles were able to predict known natural interactions, but the natural peptide ligands were generally more hydrophilic.

Comparative analysis revealed that our phage-derived ligands for PRMs often resemble peptide ligands bound to similar PRMs in known

PRM/peptide complex structures, suggesting that the natural and optimal peptides likely use similar molecular interactions to bind PRMs. In addition, six of eight phage-derived specificity profiles showed good agreement with ELM motifs, confirming that these phage-derived ligands mimic natural ligands. Notably, differences between the optimal peptides in our database and the predominantly natural peptides in the structural and ELM databases can provide valuable insights to better understand the structural basis of PRM/peptide recognition, which in turn could aid the design of peptide-based inhibitors to target PRMs in cells. In addition, our database may also prove useful for guiding peptidomimetic design and the peptides could be used as intracellular inhibitors for target discovery and validation (Qvit *et al*, 2017; Robertson & Spring, 2018).

Phage-derived ligands rarely match natural ligands exactly, mostly because of differences between *in vitro* and natural evolutionary processes. Although *in vitro* evolution is driven to maximize affinity, natural evolution is driven by the need for high specificity to reduce cross-reactivity with the thousands of non-partner proteins in the cell. Nevertheless, our database should prove useful to help the generation of new annotations for future functional ligands with optimized affinities and to guide further exploration of PRM families that have not yet been studied. Moreover, the database can be used to predict natural interaction partners and provide insights into potential cellular functions of PRMs, as has been shown previously by us and others (Schon *et al*, 2002; Slivka *et al*, 2009; Zhang *et al*, 2009; Reimand *et al*, 2012; Kelil *et al*, 2016; Hershey *et al*, 2016).

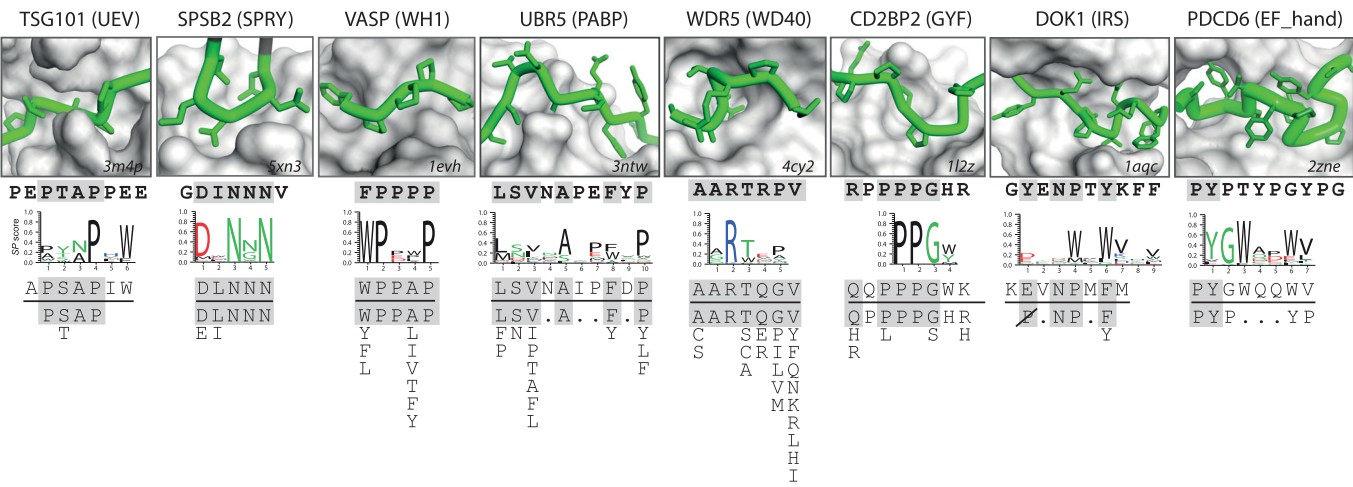

**Figure 6.  Comparison of phage-derived ligands and natural SLiMs in complex with PRMs.**

Depicted are eight PRMs for which phage-derived specificity profiles were determined (Fig 1) and for which natural SLiM ligand information is available in the ELM repository (Dataset EV6). The name of the protein from which the studied PRM was derived is listed at the top with the PRM family name in parenthesis. The box shows a close-up of the structure of the peptide-binding site of the PRM, which is depicted as a gray surface (the PDB entry code is shown in the bottom right corner). The peptide ligand is colored green with the main chain shown as a tube and side chains shown as sticks (the N terminus is to the left), and its sequence is shown directly below the structure in bold text. Below the peptide ligand sequence, the following are shown: the phage-derived specificity profile, the phage-derived peptide most similar to the ELM motif, and the ELM motif (below the horizontal line). Sequences in the peptide structure, the peptide ligand, and the phage-derived peptide that match the ELM motif are shaded gray. The ELM motifs are arranged vertically to align with the specificity profile and peptides. Position allowing any amino acid except Pro is depicted as "P" crossed out with a diagonal line.

# Materials and Methods

### Selection, expression, and purification of domains

All domains from reviewed human proteins in the UniProt database (The UniProt Consortium, 2012) were downloaded using Pfam-A v31.0 signatures (Punta *et al*, 2012). For 18,708 out of the 20,431 reviewed human proteins in UniProt, a total of 48,091 domains were identified and grouped in 6,483 unique Pfam-A families (Dataset EV1). The SCOWLP (Teyra *et al*, 2011) and 3DID (Mosca *et al*, 2014) databases were used to select all domains with structures solved in complex with a peptide, and the union of the two sets gave a total of 503 unique PRM families (Dataset EV2). A total of 285 domains representing 115 different structural families were selected for expression and purification. Domain boundaries were manually defined by aligning the domain sequence to the closest PDB structure and taking equivalent boundaries. There were 15 cases where the construct was generated from two or more domain subunits since their interface resembled a hydrophobic core and would not fold or be soluble independently (Dataset EV3). In these cases, the subunits belonged to different families, but were counted as a single domain and as a single family, which was the combination of both families. A DNA fragment encoding each domain was synthesized by commercial vendors (GenScript Inc., GeneArt and Biobasic) and was cloned into a vector designed for expression and purification of a fusion protein consisting of the domain fused to the C-terminus of glutathione *S*-transferase (GST) (Huang & Sidhu, 2011). The resulting set was arrayed in 96-well plates for high-throughput protein expression and purification, as described (Huang & Sidhu, 2011).

### Peptide-phage library construction

A diverse hexadecapeptide phage-displayed library was generated for identification of peptides binding to the PRMs. An IPTG-inducible $P_{tac}$ promoter was utilized to drive the expression of open-reading frames encoding the fusion proteins in the following form: the stII secretion signal sequence, followed by a random peptide flanked with linkers at both ends, followed by the M13 bacteriophage gene-8 major coat protein (P8). The libraries were constructed by using oligonucleotide-directed mutagenesis with the phagemid pRSTOP4 as the template, as described (Chen *et al*, 2015). The mutagenic oligonucleotides used for library construction were synthesized using a custom trimer phosphoramidite mix (TriLink BioTechnologies), containing equimolar quantities of 19 codon trimers encoding all genetically encoded amino acids except cysteine. The diversity of the library was $4 \times 10^{10}$ unique peptides.

### High-throughput peptide-phage display selections

The phage-displayed peptide library was used to select binding clones for each of the purified GST-PRM fusion proteins in a high-throughput format. The exact protocol carried out in this study has been published previously (Huang & Sidhu, 2011). Five rounds of selections were conducted to enrich peptide-phage that bound to each GST-PRM fusion protein. Approximately 48 individual clones from rounds 4 and 5 were assessed by phage ELISAs to identify clones that bound to the target protein but not to GST (Tonikian *et al*, 2007). Positive clones were sequenced to compile a set of 2,173 binding peptides, of which, 1,091 were unique (Dataset EV4).

## Generation of specificity profiles

In order to obtain the specificity profiles for each PRM with five or more binding peptides, all the motifs present in the sequences were identified and used to cluster the peptides into distinct groups of similarity using the clustering algorithm CLUSS (Kelil *et al*, 2007). The peptides within each group were aligned based on the motif, and the alignments for the different groups were merged using the profile alignment algorithm ClustalW2 (Larkin *et al*, 2007). Peptides that clustered as outliers were eliminated from the alignment, and in some cases, the alignment was manually curated to optimize the functional specificity known in the literature. The sequence alignments were converted to position weighted matrices (PWMs), where the weight (*W*) of each amino acid at each position equals its frequency multiplied by the difference between the frequencies of the most and least abundant amino acid at the same positions, as follows:

$$W_i^j = f_i^j \times \left( \max f^j - \min f^j \right),$$

where $W_i^j$ is the weight of amino acid *i* at position *j*, $f_i^j$ is the frequency of amino acid *i* at position *j* in the alignment, and $\max f^j$ and $\min f^j$ are the maximum and minimum frequencies of amino acids at the position *j* in the alignment, respectively. Thus, the smaller the difference between $\max f^j$ and $\min f^j$, the smaller the frequency of every amino acid at that position of the PWM. The specificity potential (*SP*) score at each position was defined as the sum of amino acids weights. The PWMs were visualized as sequence logos (Schneider & Stephens, 1990), where at each position the relative heights of the letters indicate their weighted frequency in the PWM, and the total height of the letters indicates the PRM specificity at that position (Fig 1). The *SP* score equals one or zero when the PRM is completely specific for a single amino acid at a given position or when there is no preferred amino acid at a given position, respectively.

In the peptide-phage library, randomized hexadecapeptides were flanked by glycine residues that can also participate in PRM recognition. Thus, in cases where logos had specific positions located at the N- or C-terminal end of the randomized region, a glycine residue was inserted to the aligned peptide sequences at the same terminal region and a new PWM was calculated. Finally, the flanking positions with low *SP* values (< 0.2) were trimmed to select the core significant region of the PWM, and the total specificity potential score (*SP*) was calculated by summing up the *SP* scores across all positions in the logo with *SP* ≥ 0.2. The same strategy was applied to previous binding peptide data for human PDZ (Tonikian *et al*, 2008), SH3 (Teyra *et al*, 2017) and WW domains (unpublished) (Appendix Fig S2).

## Characterization of physicochemical properties for peptide ligands

Disordered regions of the proteome were obtained from 15,391 reviewed, human protein sequences from UniProt (The UniProt Consortium, 2012) and defined based on the IUPred2A algorithm with a conservative 0.5 cut-off value (Mészáros *et al*, 2018). A total of 26,283 intrinsic disordered regions, including a total of 1,996,667 amino acids, were obtained and defined as the "disorderome", which was artificially fragmented to generate a total of 160,713 hexadecapeptides.

For each PRM, each position that was part of the specificity profile was classified as either specific (*SP* ≥ 0.33) or non-specific (*SP* < 0.33) (Appendix Fig S1), and the amino acid frequencies and the hydrophobicity scores were calculated. The average frequency for each amino acid at the 276 specific and 267 non-specific positions in the 74 PRM specificity profiles and within the human disorderome (1,996,667 amino acids) were calculated (Fig 2C). The Roseman's Hydropathy Index (RHI) (Roseman, 1988) was used to assign the hydrophobicity for each amino acid. The RHI for each amino acid is calculated based on the distribution coefficient of partitioning between water and non-polar octanol, which is thought to mimic the environment of a protein core, and is corrected for the solvation effects of neighboring peptide bonds at the backbone, which reduce the hydrophilicity of polar side chains (Roseman, 1988). Gly has a benchmark RHI of zero, as it lacks a side chain, and amino acids with values above or below zero are considered hydrophobic or hydrophilic, respectively (Fig 2C). The hydrophobicity scores for each of the 276 specific and 267 non-specific positions in the 74 PRM specificity profiles were calculated as the sum of all amino acid frequencies multiplied by their RHI (Roseman, 1988), and the distribution of the hydrophobicity for the positions was generated (Fig 2D). The distribution of the mean amino acid RHI for each of the 160,713 disorderome hexadecapeptides was also calculated (Fig 2D).

ELM classes, SLiM instances, and protein/SLiM interactions were downloaded from the ELM database (Version: 1.4)(Kumar *et al*, 2020). A total of 1,385 SLiMs in ligand sites were extracted from ELM instances file and mapped to the ELM class motifs to identify specific and non-specific positions, which are defined as positions that show preferences for a subset of residues and those that can tolerate any substitution (wild card "."), respectively. The amino acid frequencies and the hydrophobicity scores were calculated for all positions in the SLiMs within an ELM class, as described above, and averaged across the class to ensure no bias toward classes with large numbers of annotated instances (Fig 2C and D). In addition, a total of 8 PRMs phage-derived specificity profiles were identified in the ELM protein/SLiM interactions file based on Uniprot_id of the PRMs, and the ELM class for the PRM family was used to compare the motif to the phage-derived peptides to identify the best hit (Dataset EV6).

## Comparison with PDB structures

PSI-BLAST (5 iterations and *e*-value = $1^{-05}$) was applied to each of the 163 PRM sequences against all sequences in the PDB (www.rcsb.org), and each PDB structure containing a protein chain with a sequence alignment coverage > 75% with the corresponding PRM sequence was selected. From the selected structures, those containing at least one chain aligning with the PRM sequence and at least one other chain containing 5–30 amino acids were collected. For these structures, the similarity between the structure peptide and each phage-derived peptide was calculated. Amino acids were considered to be similar based on physicochemical relationships (Taylor, 1986), and the following groups were assigned: W, F, Y, H (aromatic); N, Q, K, R, H (large polar and basic); N, Q, D, E (large polar and acidic); N, S, T (small polar); V, I, L, M, F (large hydrophobic); A, T, V, I (small hydrophobic); G (unique); P (unique); C (unique). Amino acids with post-translational modifications were excluded from the calculations. For each PRM-ligand pair in our database, a best-matched PRM-ligand

complex from the PDB was chosen with the highest ligand similarity and a high PRM identity (Dataset EV5). The chosen structures were manually inspected to identify the portion of the peptide ligand and the residues that interacted with the PRM using Discovery Studio Visualization Software (BIOVIA).

## Data availability

The datasets and materials produced in this study are available in the following databases:

- Phage-derived data (www.prm-db.org)
- Most of PRM constructs for expression and purification: Addgene plasmid repository (https://www.addgene.org/Sachdev_Sidhu/)

**Expanded View** for this article is available online.

## Acknowledgements

We are grateful to H. Huang, R. Arnold, K. Boonen, and Y. Ivarsson for technical support. This work was supported by an operating grant from the Canadian Institutes of Health Research (MOP-93684) awarded to S.S.S.

## Author contributions

Conceptualization: MS, PMK, GB, SEN, JM, and SSS; Experiments: JT, AK, YH, AAD and JG; Analysis: JT and AK; Web-based database development: SJ, MH, and RJ; Writing, JT, AK, and SSS; Supervision, GDB and SSS; Funding Acquisition, SSS.

## Conflict of interest

The authors declare that they have no conflict of interest.

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
