## [Review Process File · Molecular Systems Biology]

Large-scale survey and database of high affinity ligands for peptide recognition modules

Joan Teyra, Abdellali Kelil, Shobhit Jain, mohamed Helmy, Raghav Jajodia, Yogesh Hooda, Jun Gu, Akshay D'Cruz, Sandra E. Nicholson, Jinrong Min, Marius Sudol, Philip Kim, Gary Bader, and Sachdev Sidhu

DOI: [10.15252/msb.20199310](https://doi.org/10.15252/msb.20199310)

Corresponding author(s): Sachdev Sidhu (sachdev.sidhu@utoronto.ca)

Review Timeline:

Submission Date:	21st Oct 19
Editorial Decision:	21st Nov 19
Revision Received:	16th Oct 20
Editorial Decision:	29th Oct 20
Revision Received:	3rd Nov 20
Accepted:	4th Nov 20

Editor: Maria Polychronidou

Transaction Report:

Thank you again for submitting your work to Molecular Systems Biology. We have now heard back from the three referees who agreed to evaluate your study. As you will see below, the reviewers acknowledge that the study presents a relevant resource. They raise however a series of concerns, which we would ask you to address in a major revision.

Without repeating all the points listed below, one of the more fundamental issues is raised by Reviewers #1 and #2, who point out that despite its potential resource value the study remains somewhat thin in terms of providing new biological insights. We think that expanding the study by following up on one of the suggestions of reviewer #1 and the recommendations of reviewer #2 would significantly enhance the impact of the study.

All other issues raised by the reviewers would need to be convincingly addressed. Please feel free to contact me in case you would like to discuss in further detail any of the issues raised.

On a more editorial level, we would ask you to address the following issues.

REFEREE REPORTS

Reviewer #1:

In this manuscript Teyra, Kelil and colleagues have used an established phage-display approach to determine peptide binders for 163 globular domains (peptide recognition modules, PRMs). In opposition to previous related studies that have focused on profiling different instances of specific domain families, this work tried to survey a diversity of different families, having determined at least one binding peptide for 79 different domain families. Out of the 163 domains, 74 had at least 5 peptides from which a specificity model could be derived. The authors briefly compared some of the properties (length, constrained positions) of these specificity profiles with those found for specific peptide binding domain families (SH3, PDZ, WW). They then matched the profiled domains with appropriate structural models, showing that the constrained positions were often in-line with the peptide bound in the domain-ligand complex. The collected information has been made available in an easy to use database. Studying liner-motif interactions remains a very difficult challenge and I think this work and the accompanying database serves as a fantastic resource for a number of potential future studies. However, the authors themselves have not really derived new knowledge from this resource, besides the determination of the binding peptides and specificity models. Providing some example application(s) would strengthen this manuscript considerably.

Major concerns

My single biggest concerns is that there is really little in this work in terms of novel biological

findings. Previous related works have tried to study, for example, the evolution of peptide binding interactions, or the extent by which the same domains may have more than one binding mode, or trying to make concrete predictions for in-vivo targets of peptide binding domains. I understand that a lot of work has gone into obtaining the domain binding peptides but this manuscript would be considerably stronger if the authors then used the data for some application akin to those prior studies. To be constructive I provide here some suggestions of potential applications. However, I don't mean that the authors should do all of these things or even any of these things, just that I suggest the authors should showcase how their new data can be used to derive new biological findings. Suggested possible applications could include:

- One of the most striking aspects of this resource is that the authors have covered many domain families and have structural models for a very large fraction of them. This aspect is not really explored. Looking at the beautiful Figure 4, it seems clear that there is a large diversity of folds and no immediate patterns relating the fold and the motifs appears but this relation between fold, pocket and binding peptides could be quantified. Is there a relation between the properties of the fold and the types of sequences it binds? Any relation between the residues near the peptide domains and the amino-acids in the binding peptides? Are the domain residues in contact with target peptide residues more likely to be conserved? Are the domain-peptide specific residue-residue contacts for constrained peptide positions more likely to be important for binding?

- In Figure 4, some of the specificity models for some domains (e.g. HSPA9, SND1, WDR74, NXF1, others) extend beyond the size of the structural ligands. Some of these extensions still contain positions that are apparently constrained. Can the authors use the structural models to rationalize these extended specificities? For some of these examples, the extended binding mode may already be described in the literature but any new examples could be interesting.

- Perhaps a low hanging fruit application would to suggest likely in-vivo protein binding partners and the binding sites for these domains. The authors have already started to do some of this work by predicting likely binding sites within disordered regions. These predictions could be overlaid on an up-to-date compilation of human interaction data and indicating potential binding site regions that could contribute to the protein-protein interactions. Providing some examples from this could help others understand how to make use of this resource. A larger extension of this could be to map and study human genetic variation on to these potential binding sites.

Minor concerns

- In Figure 4, the authors note that there are residues that are modified by phosphorylation and other PTMs. However, this is not really discussed in the results. I found it interesting that these positions provided clear mismatches between the phage-display selected amino-acids and the amino-acid in the structural ligand. For phosphorylation these are the expected S/T to phospho-mimetic D/E mismatches but I am not familiar with the idea that tryptophan can mimic methylated residues. This is most apparent for the TUDOR domain of SND1 with a strong selection for tryptophan at the position selective for methylated arginine. Is this well known? It would be worth having a short section describing in more detail the results for the PTM binding domains, ideally looking in detail at the structural reasons for some of the differences. Even within the phosphorylation examples there are interesting differences that are worth going into more detail. For example, the two first 14-3-3 domains in Figure 4 have a phospho-threonine and phospho-serine in the structural peptide and they select different mimetic residues and the two phosphotyrosine bound domains (DOK4 and SHC1) don't really appear to select for phospho-mimetic residues at those positions.

- Figure 4 is beautiful but maybe too big for a paper. The authors could consider splitting into more

than one figure. If they were to discuss the PTM bound structures they could move those onto a separate figure.

- From 215 domains that the authors could express they could confidently determine a binding peptide for 163 and 74 had 5 or more peptides bound. Given the diversity of domains selected it could be that some of the domains are clearly peptide binding domains and some may bind peptides weakly at an interface that is most often binding another type of molecule. I wonder if this would be apparent when the drop-off rate is studied in aggregate. Is there an enrichment for domains that are also known to bind other types of molecules in those that did not bind any or many peptides ?

Reviewer #2:

Teyra et al. performed phage display screening to identify peptide binders of a large number of protein recognition modules (PRMs). They screened a 16-mer amino acid library against 163 PRMs and validated many peptides from rounds 4 and 5 of screening using phage ELISA. For many PRMs they validated <5 peptide binders this way. For 74 PRMs with >5 validated peptide binders that showed residue preferences at specific sites, they used the hits to define PRM specificity profiles and build position weight matrices (PWMs). The authors also identified possible binding models for their hits by finding the PDB structure with closest sequence similarity between the domain and ligand pair with a hit from the phage screening. The PWMs were also used to evaluate the human proteome and identify high-scoring sequences. The data generated are freely available for download, and the authors built a web site for easy access. The authors state that clones have been deposited at AddGene (though I don't see them listed there yet). This large-scale study presents a tremendous investment of resources that is out of the reach of most academic-scale labs; the results should certainly be published and made widely available.

The paper does not really provide any key new insights into PRM specificity. The results consist of reporting what was found, and summarizing the data using plots that compare the length, composition and specificity of the motifs defined here and those defined previously for SH3, SH2 and WW domains using a similar approach. A major emphasis is on comparing phage hits to examples of peptide-binding structures in the PDB, arguing that sequence similarities support binding of phage-derived peptide in the same mode (or at the same site, such that they would function as competitive inhibitors), though this is subjective claim without much support.

To put this work in an appropriate context, these points should be addressed in the manuscript:

- A major point is that screening for the tightest-binding (or most slowly dissociating) peptides for a PRM by phage display gives hits that differ from what that PRM binds in a biological context. The authors choose to emphasize similarities rather than differences between their hits and native ligands, but this is misleading and could easily lead to mis-use of the reported data. This is particularly true because the authors present matches to their phage-derived PWM as "Predicted Human Interactions" in their on-line database, which (I think) cannot be what they actually mean, in the absence of filtering to suggest that interaction in a cellular context is plausible. These screening data are what they are, and should be presented as such. If the authors want to argue that these data have value for identifying biological interaction motif instances, they need to argue that more convincingly rather than implying it without evidence.

- The most striking result from screening is the extremely high prevalence of Trp (20% of residues at "specific" positions). The authors suggest this is similar to what has been shown for human SLiM, but I think that is an overstatement. A realistic estimate of the extent to which Trp is enriched in this screen vs. in other determinations of PRM-binding peptides should be included in this paper.

- For many of these PRMs, a good deal is known about their specificities from prior work. For some

domains, analysis of native partner interactions has provided a different impression of the specificity profile than the phage results. This needs to be acknowledged, with reference to ELM or other databases that compile what we know so far about PRM binding specificity data. The authors should present at least one example of a case where the phage results give a different picture than what was already reported. In this vein, it would be helpful if the authors could indicate for which domains there was vs. wasn't prior information about the PRM specificity. Ideally, comparisons to the disorderome would also include comparisons to the "SLiM-ome" (or the union of all reported SLiMs) as defined by ELM. Addressing these things would help clarify the novelty of the authors' findings.

- In many cases is not convincing that the phage-derived peptides derived from the screen are binding in the same mode as many of the structures the authors compare to. Figure 4 is presented as evidence, but similarities highlighted in this figure are subjective and are sometime not very high even by this loose standard (examples: ARIID4A, TDRD1, VCL, etc.). Calling out the similarities and downplaying the differences is misleading. Much more could be done to bolster confidence in a subset of the interactions (e.g. using comparisons to motifs already defined in the literature), though we acknowledge that this may be beyond the scope of this report.

- The experimental methods are described predominantly by referencing earlier work. It is difficult to be confident that these earlier papers provide a complete description of what was done, e.g. of the phage screening and ELISA protocols. This is particularly true because different papers are cited in the text vs. method sections (e.g. Teyra 2017 vs. Huang & Sidhu 2011) Were experimental variables such as concentrations, buffers, and wash stringencies exactly as in these earlier works? It would be good to associate this large data set unambiguously with the protocol(s) that generated it.

- The organization of the article could be improved. The text is redundant in places. Also, material suitable for the Introduction or Discussion appears in the main text, and aspects of the methods appear in the main text but not the methods section. Some of the supplementary materials have extra sheets in the excel file that don't appear intended for release. The authors should go through all supplementary tables and ensure that the terms used in the headings are precisely defined, with an equation if appropriate (all scores or ratios that are reported should be rigorously defined). The paper reads as if it were rather hastily written.

The authors have the wrong year in this citation: Davey NE, Roey KV, Weatheritt RJ, Toedt G, Uyar B, Altenberg B, Budd A, Diella F, Dinkel H & Gibson TJ (2011) Attributes of short linear motifs. *Mol. Biosyst.* 8: 268-281

Reviewer #3:

This manuscript reports a large scale survey of peptide ligands by phage display with many different protein domains, together with a derived database where users can explore the data. Now that it is clear that a huge amount of cell regulation involves short linear motifs (SLiMs), medium and high throughput methods are essential to screen for candidate motifs that can be subjected to low throughput validation. Phage display is an essential tool for this endeavour. The resource prm-db built upon the experimental work presented here will be explored by (increasing numbers of) experimentalists seeking for motif interactors in their systems. It might also prove useful for annotated motif resources such as ELM to help in motif refinement, especially when only a couple of true instances are known. Overall, this is valuable work for the SLiM field.

One issue with phage display is that there can be a hydrophobic/Tryptophan bias in the recovered peptides. Importantly, this is addressed nicely in the discussion. There certainly are plenty of W-containing SLiMs such as DPW and WRPW and it is true that W is present more often than would be expected by chance. And many of the phage display motifs in the resource don't have W

enrichment so then there is no issue. It is good, though, that researchers are properly informed.

I have no revisions to request.

Point-by-point Response to reviewers:**Reviewer #1:****Comments:**

In this manuscript Teyra, Kelil and colleagues have used an established phage-display approach to determine peptide binders for 163 globular domains (peptide recognition modules, PRMs). In opposition to previous related studies that have focused on profiling different instances of specific domain families, this work tried to survey a diversity of different families, having determined at least one binding peptide for 79 different domain families. Out of the 163 domains, 74 had at least 5 peptides from which a specificity model could be derived.

The authors briefly compared some of the properties (length, constrained positions) of these specificity profiles with those found for specific peptide binding domain families (SH3, PDZ, WW). They then matched the profiled domains with appropriate structural models, showing that the constrained positions were often in-line with the peptide bound in the domain-ligand complex. The collected information has been made available in an easy to use database.

Studying liner-motif interactions remains a very difficult challenge and I think this work and the accompanying database serves as a fantastic resource for a number of potential future studies. However, the authors themselves have not really derived new knowledge from this resource, besides the determination of the binding peptides and specificity models. Providing some example application(s) would strengthen this manuscript considerably.

Major concerns: My single biggest concern is that there is really little in this work in terms of novel biological findings. Previous related works have tried to study, for example, the evolution of peptide binding interactions, or the extent by which the same domains may have more than one binding mode, or trying to make concrete predictions for in-vivo targets of peptide binding domains.

I understand that a lot of work has gone into obtaining the domain binding peptides but this manuscript would be considerably stronger if the authors then used the data for some application akin to those prior studies. To be constructive I provide here some suggestions of potential applications. However, I don't mean that the authors should do all of these things or even any of these things, just that I suggest the authors should showcase how their new data can be used to derive new biological findings. Suggested possible applications could include:

- One of the most striking aspects of this resource is that the authors have covered many domain families and have structural models for a very large fraction of them. This aspect is not really explored. Looking at the beautiful Figure 4, it seems clear that there is a large diversity of folds and no immediate patterns relating the fold and the motifs appears but this relation between fold, pocket and binding peptides could be quantified. Is there a relation between the properties of the fold and the types of sequences it binds? Any relation between the residues near the peptide domains and the amino-acids in the binding peptides? Are the domain residues in contact with target peptide residues more likely to be conserved? Are the domain-peptide specific residue-residue contacts for constrained peptide positions more likely to be important for binding?

- In Figure 4, some of the specificity models for some domains (e.g. HSPA9, SND1, WDR74, NXF1, others) extend beyond the size of the structural ligands. Some of these extensions still contain positions that are apparently constrained. Can the authors use the structural models to rationalize these extended specificities? For some of these examples, the extended binding mode may already be described in the literature but any new examples could be interesting.

- Perhaps a low hanging fruit application would to suggest likely in-vivo protein binding partners and the binding sites for these domains. The authors have already started to do some of this work by predicting likely binding sites within disordered regions. These predictions could be overlaid on an up-to-date compilation of human interaction data and indicating potential binding site regions that could contribute to the protein-protein interactions. Providing some examples from this could help others understand how to make use of this resource. A larger extension of this could be to map and study human genetic variation on to these potential binding sites.

Response:

Thank you for the positive comments and constructive criticisms. Our priority has been to try to get our data out to the community as fast as possible so that many different groups can start to derive biological insight from these data and we submitted it as a Research Resource rather than an Article. However, we agree that it would be useful to delve more into the details of our data.

Figure 4 is a panel of known PRM/peptide structures that are more similar to our PRM specificity profiles. Any effort on analyzing residue-residue interactions or to understand the functional role of extension in some of the specificity profiles would require a massive structural modelling and molecular dynamics. In addition, we believe the field is not accurate enough to

obtain reliable structural models to perform analysis at atomic detail. In addition, any conclusion obtained from modelling would still require structural biology efforts for validation. We may pursue some of these studies in the future with collaborators, but most importantly, making this complete database available online will enable many other structural biologists to engage in studies of this kind, and importantly, researchers with existing expertise and interest in particular PRM families will be best equipped and motivated to use our database to enable studies of key PRMs of particular interest.

Similarly, prediction of protein binding partners for the 74 PRMs using the specificity profiles would generate massive lists of interactions, but without validation of the novel binders it would not yield new biological insights. Even GO terms enrichment analysis of the predicted binders would obtain generic terms that would not contribute on deriving biological findings. Moreover, we and others have noted (including in the current paper) that the phage-derived motifs do not match perfectly to natural ligands because phage methods optimize affinity independent of biological context. Thus, we believe that the type of data we have generated is best used by experts of particular PRMs in conjunction with other data to derive putative natural partners that can then be explored in depth by groups with existing expertise in particular areas of biology. As with the structural studies described above, we feel strongly that these goals will be best achieved by the timely release of our complete database to the entire research community to enable detailed studies of this kind.

We appreciate that the reviewer did not insist on the studies he/she proposed, but rather, gave these as some examples of detailed studies we could pursue to strengthen our work. Considering our interests and expertise, together with the data on hand, we decided to strengthen our manuscript by analyzing the biological significance of our phage-derived peptides and specificity profiles in comparison with the functional SLiMs and the motifs from ELM database at two levels:

- An addition to the section “General features of PRM specificity profiles”, which includes a comparison of the amino acid propensities and physicochemical properties of our ligands with the 1,385 functional SLiMs in ligand sites contained in ELM database. This section shows that the characteristics of phage-derived peptides partially reflect those from functional SLiMs, and the differences are explained.*
- A new section “Comparison of phage-derived ligands with functional ligands” contains comparative analysis of all phage ligands to the ELM class motifs to assess their agreement. We identified only 8 PRMs with SLiM interacting information from the ELM*

database and with enough phage-derived peptides to generate specificity profiles. We analyzed these 8 high resolution cases in detail. Importantly, we observed very good agreement for 6 of 8 examples.

We believe that these extensive additions of detailed analyses address the reviewers request for additional studies, and they position our phage-derived data in comparison with natural interactions in a manner that is most sensible, specifically, a comparison of our short peptides to short natural peptides and motifs in existing databases. We are encouraged that this analysis revealed good agreement between the phage-derived and natural ligands, along with some differences which are explicable by the different contexts of the datasets.

Minor concerns

- In Figure 4, the authors note that there are residues that are modified by phosphorylation and other PTMs. However, this is not really discussed in the results. I found it interesting that these positions provided clear mismatches between the phage-display selected amino-acids and the amino-acid in the structural ligand. For phosphorylation these are the expected S/T to phospho-mimetic D/E mismatches but I am not familiar with the idea that tryptophan can mimic methylated residues. This is most apparent for the TUDOR domain of SND1 with a strong selection for tryptophan at the position selective for methylated arginine. Is this well known? It would be worth having a short section describing in more detail the results for the PTM binding domains, ideally looking in detail at the structural reasons for some of the differences. Even within the phosphorylation examples there are interesting differences that are worth going into more detail. For example, the two first 14-3-3 domains in Figure 4 have a phospho-threonine and phospho-serine in the structural peptide and they select different mimetic residues and the two phospho-tyrosine bound domains (DOK4 and SHC1) don't really appear to select for phospho-mimetic residues at those positions.

Response:

We agree with the reviewer that the specific cases of PRMs that naturally recognize PTMs are worthy of more detailed discussion. Consequently, an extensive new section entitled “Phage-derived mimics of peptides containing PTMs” and a new Figure 5 have been included in the results to discuss the 17 cases where a structural peptide contains a PTM amino acid and the different mimetic residues observed in our data. This new section reads as follows:

“The 17 structure peptides with PTMs were divided into three groups of eight, four or five peptides containing phosphorylated serine/threonine (pSer/pThr, **Fig 5A**), phosphorylated tyrosine (pTyr, **Fig 5B**) or methylated Arg/Lys (meArg/meLys, **Fig 5C**), respectively. Six of the eight PRMs that recognized pSer/pThr belonged to the 14-3-3 domain family and the other two belonged to the CKS or NIF domain family. In five of these, the aligned phage-derived peptide contained a negatively-charged Asp/Glu residue in place of the pSer/pThr residue in the structure peptide, consistent with other reports that have shown that Asp/Glu can effectively mimic the shape and charge of pSer/pThr (Sundell et al, 2018). For two of the other PRMs, the 14-3-3 domains of YWHA E and YWHA Z, the phage-derived peptide contained an aromatic Tyr/Trp residue in place of pSer/pThr. The structure peptide for the remaining PRM, the NIF domain of CTDSP2, was unusual in that it contained two pSer residues and exhibited only minimal homology with the phage-derived peptide, thus making it unclear whether the phage-derived peptide bound to the same site as the structure peptide. Three of the four PRMs that recognized pTyr belonged to the IRS domain family and the fourth belonged to the PID family, and in each case, the alignment showed that the phage-derived peptide contained a hydrophobic residue in place of the pTyr in the structure peptide. Finally, the five PRMs that recognized meArg/meLys included two TUDOR domains, a PHD domain, a TUDOR-knot domain, and a WD40 domain. Except for the WD40 domain, the alignments revealed that each phage-derived peptide substituted a hydrophobic residue for the meArg/meLys residue in the structure peptide. In the case of the WD40 domain, meLys in the structure peptide was substituted by His in the phage-derived peptide, but in this case, the structure peptide showed low similarity with the phage-derived peptide and specificity logo, making it uncertain whether the two peptides recognize the same site in the same manner. Taken together, these results showed that phage-derived peptides without PTMs can mimic peptide ligands that contain PTMs in many cases, either by using Asp/Glu residues that mimic pSer/pThr residues or by using hydrophobic residues that likely act as partial mimics of PTMs. Thus, our results could be useful for designing PTM mimics, but further biophysical and structural studies will be necessary to reveal the molecular basis for PTM mimicry.”

We believe that this aspect of our work is worthy of additional structural and functional studies which will likely be forthcoming soon. Again, release of our online database to the community will enable other researchers with expertise in these areas to pursue these studies also.

Comment:

Figure 4 is beautiful but maybe too big for a paper. The authors could consider splitting into more than one figure. If they were to discuss the PTM bound structures they could move those onto a separate figure.

Response:

We agree, and following the reviewer's suggestion, we have moved the PTM-containing peptide structures to a new Figure 5 that includes cases with and without specificity profiles.

Comment:

From 215 domains that the authors could express they could confidently determine a binding peptide for 163 and 74 had 5 or more peptides bound. Given the diversity of domains selected it could be that some of the domains are clearly peptide binding domains and some may bind peptides weakly at an interface that is most often binding another type of molecule. I wonder if this would be apparent when the drop-off rate is studied in aggregate. Is there an enrichment for domains that are also known to bind other types of molecules in those that did not bind any or many peptides?

Response:

We agree with the reviewer that peptide-phage display may not have worked for some PRMs because they bind peptides too weakly to be detected by phage-display technology, although other reasons might exist. Note that there are many families where we tested multiple PRMs and were able to obtain results for a subset of them (i.e. 5/14 IRS, 5/9 VHS, or 4/10 SPRY domains), suggesting that the variability may also be explained by purification and folding differences between the members, since family members are expected to recognize peptides in a comparable fashion. We added a new paragraph in the results section "A catalog of PRM binding specificities" that explains possible reasons of purification and peptide identification failures: "Failure to purify 70 PRMs may be due to non-optimal boundaries for the expression constructs or instability of the domains in isolation. Failure to identify binding peptides may be due to weak affinities that cannot be selected by the phage display method or specificity for post-translational modifications (PTMs) that cannot be mimicked by standard amino acids."

We also followed reviewer's suggestion and studied if the 74 PRMs with logos (multiple peptides selected) were more or less prone to interact with other domains than those PRMs where we did not obtain any or enough peptides to generate a logo. Thus, we downloaded the

PRM-peptide and PRM-domain interaction data from the 3DID structural database and organized the 285 putative PRMs into three **PRM binding groups**: “only peptides” (structures of members of the PRM family binding only to peptide ligands), “only domains” (structures of members of the PRM family binding only to other domains) or “both peptides and domains” (some structures of members of the PRM family bind to peptides and others bind to domains) . Then, we calculated for each group the frequencies for each of the three **PRM types**: “domains with logos” (bound to enough phage-derived peptides to obtain logos), “domains with no logos” (bound to phage-derived peptides but not enough to obtain logos), and “domains with no peptides” (did not bind to phage-derived peptides) (see table below). Our results show that “domains with logos” are mostly known to bind to “only domains” or to “both domains and peptides” (37.8% and 37.8%), showing that the “domains with logos” (29.7%) are not clearly peptide binding domains as the reviewer hypothesized.

To evaluate if the differences among **PRM type** were significant, we compared their frequencies to those of the total 285 PRMs per **PRM binding group** using the cumulative hypergeometric distribution, which scores the probability to see by chance at least k successes in a sample of size n picked from a finite population of size N containing m successes. As shown in the table below, we found weak p -values (all well above 0.001) which demonstrate that there is no significant difference in the frequencies of the **PRM types** for any of **the PRM binding group**. Thus, the answer to the reviewer’s question is that there is no enrichment for domains that are also known to bind other types of molecules in those that did not bind any or many peptides.

We did not include this analysis in the paper because the results were as expected, and this analysis focuses on a minor unsuccessful aspect of our work (i.e. potential reasons for failed selections). However, we hope that by including it here and agreeing to the open access publication of the review response, that it will be available for those interested.

Table. Frequency and p-value of each PRM binding group by PRM type. PRM binding groups are divided in Peptide, Domain and Both (peptide and domains) based on the complex structures known for the PRM family. PRM types are divided in With logos, No logos, No peptides based on the number of ligands obtained. The p -values are calculated from the observed frequencies for each PRM types over the expected total frequencies for each PRM binding group.

		PRM binding group (% , p -value)					
PRM type	# PRMs	Peptide		Domain		Both	
		%	p -value	%	p -value	%	p -value
With logos	74	23.0%	0.56	37.8%	0.55	37.8%	0.55
No logos	89	18.4%	0.93	38.0%	0.51	42.9%	0.15
No peptides	122	11.5%	0.99	42.6%	0.06	45.9%	0.07
Total	285	15.4%		40.0%		44.2%	

Reviewer #2:

Comments:

Teyra et al. performed phage display screening to identify peptide binders of a large number of protein recognition modules (PRMs). They screened a 16-mer amino acid library against 163 PRMs and validated many peptides from rounds 4 and 5 of screening using phage ELISA. For many PRMs they validated <5 peptide binders this way. For 74 PRMs with >5 validated peptide binders that showed residue preferences at specific sites, they used the hits to define PRM specificity profiles and build position weight matrices (PWMs). The authors also identified possible binding models for their hits by finding the PDB structure with closest sequence similarity between the domain and ligand pair with a hit from the phage screening. The PWMs were also used to evaluate the human proteome and identify high-scoring sequences. The data generated are freely available for download, and the authors built a web site for easy access.

The authors state that clones have been deposited at AddGene (though I don't see them listed there yet). This large-scale study presents a tremendous investment of resources that is out of the reach of most academic-scale labs; the results should certainly be published and made widely available.

Response:

We thank the reviewer for the careful review of our work and the positive comments. We are grateful that the reviewer recognizes the value of our large-scale study and freely accessible online database and recognizes the need to publish and make the data widely available. With regards to the protein expression plasmids themselves, we also intend to make these all freely available as another valuable resource for the research community. AddGene laboratory is in the process of changing its physical location, and unfortunately, it is producing major delays in

the processing time for all submitted samples. The plasmids were sent a year ago, they have already been stored in the AddGene repository, and the deposit agreement has been completed, as shown in the attached file provided by AddGene. The link provided in the paper will be the place to look for the information once the plasmids are fully processed by AddGene.

Comments:

The paper does not really provide any key new insights into PRM specificity. The results consist of reporting what was found, and summarizing the data using plots that compare the length, composition and specificity of the motifs defined here and those defined previously for SH3, SH2 and WW domains using a similar approach. A major emphasis is on comparing phage hits to examples of peptide-binding structures in the PDB, arguing that sequence similarities support binding of phage-derived peptide in the same mode (or at the same site, such that they would function as competitive inhibitors), though this is subjective claim without much support. To put this work in an appropriate context, these points should be addressed in the manuscript:

A major point is that screening for the tightest-binding (or most slowly dissociating) peptides for a PRM by phage display gives hits that differ from what that PRM binds in a biological context. The authors choose to emphasize similarities rather than differences between their hits and native ligands, but this is misleading and could easily lead to mis-use of the reported data.

This is particularly true because the authors present matches to their phage-derived PWM as "Predicted Human Interactions" in their on-line database, which (I think) cannot be what they actually mean, in the absence of filtering to suggest that interaction in a cellular context is plausible. These screening data are what they are, and should be presented as such. If the authors want to argue that these data have value for identifying biological interaction motif instances, they need to argue that more convincingly rather than implying it without evidence.

Response:

As the reviewer points out, we present protein matches to our phage-derived PWM as "Predicted Human Interactions" in our on-line database, but we did not aim to imply that interaction in a cellular context was plausible without further filtering our results. This was explicitly explained in the "Online database of phage-derived PRM specificity profiles" section, which reads: "The database provides a list of human proteins matching PRM binding motifs, which was generated by scanning the human proteome with the PWMs. Combined with other experimental data, these can help to prioritize biological experiments to explore putative natural

interactions (Jain & Bader, 2016).” *However, we understand and agree with the reviewer that the on-line database page presenting the results of scanning the 74 PWMs against the human proteins should not be named “Predicted Human Interactions”, as it could lead to mis-use of the reported data. For clarity, we renamed it “PWM pattern matches” and we explain the calculation of the scores below the online table containing the results.*

We would like to point out that we are fully aware that phage display gives optimal ligands and this is rarely exactly the same as biological ligands which are sub-optimal for affinity. This is now noted at several points in the manuscript, as follows:

Last paragraph of the “General features of PRM specificity profiles”: “Overall, phage-derived peptides reflect the hydrophobic characteristics of functional SLiMs at specific positions, which are critical for PRM recognition”.

Last paragraph of the “Structural rationalization of PRM-ligand interactions”: “these results suggest that the phage-derived peptides for most of the PRMs in our database likely represent ligands that bind to functional sites identified previously in related PRM structures in the PDB. Differences between optimal phage-derived peptides and natural structure peptides may also be of interest to understand the types of substitutions that can enhance the affinities of natural ligands and could thus be useful for inhibitor design. Consequently, our phage-derived peptides can provide molecular insights into natural protein function and can be used as inhibitors of natural protein-protein interactions.”

Two paragraphs of the Discussion: “Comparative analysis revealed that our phage-derived ligands for PRMs often resemble peptide ligands bound to similar PRMs in known PRM/peptide complex structures, suggesting that the natural and optimal peptides likely use similar molecular interactions to bind PRMs. Notably, differences between the optimal peptides in our database and the predominantly natural peptides in the structural database can provide valuable insights to better understand the structural basis of PRM/peptide recognition, which in turn could aid the design of peptide-based inhibitors to target PRMs in cells.”

“Phage-derived ligands rarely match natural ligands exactly, mostly because of differences between in vitro and natural evolutionary processes. Whereas in vitro evolution is driven to maximize affinity, natural evolution is driven by the need for high specificity to reduce cross-reactivity with the thousands of non-partner proteins in the cell”

Comment:

The most striking result from screening is the extremely high prevalence of Trp (20% of residues at "specific" positions). The authors suggest this is similar to what has been shown for human SLiM, but I think that is an overstatement. A realistic estimate of the extent to which Trp is enriched in this screen vs. in other determinations of PRM-binding peptides should be included in this paper.

Response:

In the revised version of the manuscript, we calculated the amino acid propensities and hydrophobicity for all functional human SLiMs in the ELM database and added the information in Figure 2C and 2D. We compared this information with our results in specific and non-specific regions in our PRM specificity profiles and presented the results in the "General features of PRM specificity profiles" section, in which we compared amino acid propensities and physicochemical properties of our ligands with the 1,385 functional SLiMs in ligand sites contained in ELM database, as follows: "We also compared the characteristics of the phage-derived peptides with the disorderome and with human SLiMs from the Eukaryotic Linear Motif (ELM) repository (Kumar et al, 2020), which contains manually curated information for experimentally validated natural SLiMs and reflect natural PRM binding preferences. For better comparison with our results, we differentiated between specific and non-specific positions in the SLiMs, defined as positions that show preferences for a subset of residues and those that can tolerate any substitution, respectively."

This section shows that the characteristics of phage-derived peptides partially reflect those from functional SLiMs, and the differences are also explained:

- "A similar profile for hydrophobic residues was observed for specific positions in SLiMs, although the prevalence of Leu and Phe was 1.9- or 1.5-fold higher, respectively, compared with phage-derived ligands"
- "The second most frequent amino acid at specific positions was Pro (11%), and Pro residues were also most abundant in specific positions of SLiMs (19%) and were highly prevalent in the disorderome (13%) (**Fig 2C**)."
- "SLiMs also showed lower abundance of hydrophilic residues with a preference for charged residues in the specific positions, although the preference for negatively-charged residues over positively-charged residues was only observed in the non-specific positions (**Fig 2C**)."

- “Moreover, our analysis showed that, overall, SLiMs are much more hydrophilic than phage-derived peptides (mean RHI = -0.65 and 0.03, respectively) and slightly less hydrophilic than the disorderome (mean RHI = -0.84). However, specific positions of SLiMs are more hydrophobic than those of phage-derived ligands (mean RHI = 0.78 and 0.21, respectively), due to high prevalence of Phe, Leu and Pro in SLiMs. In contrast, non-specific positions of SLiMs are much more hydrophilic than those of phage-derived ligands (mean RHI = -2.1 and -0.23, respectively). Overall, phage-derived peptides reflect the hydrophobic character of SLiMs at specific positions, which are critical for PRM recognition.”

In addition, a new paragraph has been added, in which a discussion on the high prevalence of Trp values have been included with examples: “By far, Trp was the most frequent amino acid at specific positions in our data (20%) (Fig 2C), which agrees with what has been shown for human SLiMs (Davey et al, 2012), for which our calculations show an abundance of 8.8%. Differences might be attributed to the fact that the ELM repository does not contain SLiM instances for most of the families that recognize Trp-containing peptides (e.g. CAP-GLY, Chromo Shadow, Glycolitic, PH, SWIB, and VHS families, Fig 1). This high frequency is even more striking, considering the very low abundance of Trp in the disorderome (0.4%) (Fig 2C). Aromatic Trp side chains are often buried at interfaces, where the indole ring can form stacking interactions with other aromatic residues and cation- π interactions with Arg side chains, and the nitrogen group in the indole ring can form hydrogen bonds with polar residues (Betts & Russell, 2003).”

Comment:

For many of these PRMs, a good deal is known about their specificities from prior work. For some domains, analysis of native partner interactions has provided a different impression of the specificity profile than the phage results. This needs to be acknowledged, with reference to ELM or other databases that compile what we know so far about PRM binding specificity data. The authors should present at least one example of a case where the phage results give a different picture than what was already reported. In this vein, it would be helpful if the authors could indicate for which domains there was vs. wasn't prior information about the PRM specificity. Ideally, comparisons to the disorderome would also include comparisons to the "SLiM-ome" (or the union or all reported SLiMs) as defined by ELM. Addressing these things would help clarify the novelty of the authors' findings.

Response:

We decided to consider the reviewer's suggestion to strengthen our manuscript by comparing our phage-derived peptides and specificity profiles to the functional SLiMs and the motifs from ELM database in a new section "Comparison of phage-derived ligands with functional ligands".

*First, this section indicates which domains contain SLiM instances with PRM binding information in ELM and select the high resolution cases for further analysis, as follows: "Of the 500 PRMs in our database and the 163 PRMs in this study, we found that only 44 or 20, respectively, had at least one SLiM ligand, showing low coverage of our PRMs in the ELM repository (**Dataset EV6**). In order to compare the highest resolution examples of ELM and phage-derived results, we focused our analysis on PRMs for which our database contained enough phage-derived peptides to generate specificity profiles and for which the ELM class associated with the SLiM ligands did not contain any PTMs. For these eight out of 20 PRMs, we compared the ELM motif with the most similar phage-derived peptide and with the specificity profile, and we rationalized the comparisons using PRM structures with bound peptides (**Fig 6**)."*

*Finally, we carry out a structure/function analysis summarized in Figure 6 and explained in the section for each of the 8 cases that we could match with SLiMs. This section is described in the main text as follows: "For six PRMs - TSG101(UEV), SPSB2(SPRY), VASP(WH1), UBR5(PABP), WDR5(WD40) and CD2BP2(GYF) - the ELM motifs agreed closely with their respective phage-derived specificity profiles and with the structures of PRM-ligand complexes (**Fig 6**). The TSG101(UEV) structure contains a groove that recognizes the Pro-Thr-Ala-Pro sequence of the ligand, and Pro⁴ is the most buried residue, and also, the most conserved sequence in the specificity profile. The SPSB2(SPRY) structure contains a hydrophilic pocket that recognizes a peptide loop representing the ELM motif. Notably, positions 2 and 4 are the most solvent accessible in the structure and also the least conserved in the specificity profile, but nonetheless, several phage-derived peptides matched the ELM motif exactly. In the VASP(WH1) structure, three hydrophobic pockets are occupied by Phe¹, Pro² and Pro⁵ residues in the peptide ligand, and these residues are very similar to the ELM motif and the phage-derived specificity profile, which contain Trp¹, Pro² and Pro⁵. The Pro residues at positions 3 and 4 of the structure peptide are exposed to solvent, consistent with lower conservation of these positions in the phage-derived specificity profile. The UBR5(PABP) structure reveals an extended binding site that makes contacts with six residues imbedded within a 10-residue stretch of the peptide ligand. The importance of these six contact positions is reflected in the ELM motif, and also in the phage-derived peptides and specificity profile, which all show good agreement. The WDR5(WD40) structure contains a deep cavity that accommodates an Arg*

residue at position 2, which is completely conserved in the ELM motif and the phage-derived specificity profile. The ELM motif extends across seven positions and it closely matches the structure peptide and the phage-derived specificity profile. Finally, the CD2BP2(GYF) structure contains a hydrophobic cleft that interacts with a central Pro-Pro-Gly sequence in the peptide ligand as well as flanking residues on both ends. The short phage-derived specificity profile shows strong conservation for a PPG sequence followed by an aromatic residue, and notably, a previous study with phage display and peptide arrays defined a very similar tetrapeptide specificity profile that was validated by proteomic experiments (Kofler et al, 2005). Consequently, the short phage-derived specificity profile closely matches the core of the structure peptide and ELM motif, suggesting that this central region is most important for binding.”

Comment:

In many cases is not convincing that the phage-derived peptides derived from the screen are binding in the same mode as many of the structures the authors compare to. Figure 4 is presented as evidence, but similarities highlighted in this figure are subjective and are sometime not very high even by this loose standard (examples: ARIID4A, TDRD1, VCL, etc.). Calling out the similarities and downplaying the differences is misleading. Much more could be done to bolster confidence in a subset of the interactions (e.g. using comparisons to motifs already defined in the literature), though we acknowledge that this may be beyond the scope of this report.

Response:

The aim of the “Structural rationalization of PRM-ligand interactions” section was to figure out if the phage-derived peptides were binding to the structurally-known functional sites of the PRMs based on their similarities to the structure peptides. Our results show that 87% of the phage-derived peptides (118 of 135) exhibited >40% similarity with their corresponding structure peptides (Fig. 3B). We believe that 40% similarity should be a significant cut-off for similarity taking into account that only a subset of residues in the peptide are directly involved in PRM recognition, and that phage-derived ligands rarely match natural ligands exactly for the reasons pointed out in the Discussion section, as follows: “Whereas in vitro evolution is driven to maximize affinity, natural evolution is also driven by the need for high specificity to avoid interactions with the thousands of proteins that exist in a cell”. In addition, we identified higher similarity between phage-derived and structure peptides in positions that aligned with specific

positions in the logo than in the non-specific positions (73% vs 50%, respectively), and in positions that aligned with interacting and non-interacting positions (75% vs 29%, respectively). Overall, we believe that these results generated from **Figure 4** strongly suggest that the majority of phage-derived peptides binding in the functional binding site shown in the structure.

We agree with the reviewer that dissimilarities between phage and structure peptides are of great importance, but without the experimental biophysical or structural information of the phage peptides, we cannot assess the relevance of the amino acid substitutions at those positions. We included this point in the last paragraph of the “Structural rationalization of PRM-ligand interactions” section, as follows: “Differences between optimal phage-derived peptides and natural structure peptides may also be of interest to understand the types of substitutions that can enhance the affinities of natural ligands and could thus be useful for inhibitor design”.

The reviewer points out to three particular cases in which the reviewer believes that similarities are subjective and not very high: VCL, TDRD1 and ARIID4A. In the first case, VCL phage and structure peptides show a 75% sequence similarity, 5 of the 6 similar positions interact directly with the PRM, and the two specific positions in the logo show also similarities with the structure peptide. In the second, TDRD1 phage and structure peptides show a 66% sequence similarity and all similar positions interact to the PRM and are specific in the logo. We believe that the phage-derived peptides in these two cases are quite likely to bind in the same site shown in the structure. In the last case, ARIID4A is compared to a structure peptide with a methylated Lys and show a 50% similarity when the PTM position is not considered. This case is now discussed in the “Phage-derived mimics of peptides containing PTMs” new section in the basis of the PTM, and since the mLys to Trp mimicry has never been observed it is probable that the phage peptide might bind in a different mode or site than the structure peptide.

Overall, we find good agreement between our database and the structure database. However, we again stress that we are submitting the entire database as a resource, so all of our analysis and conclusions will be available to the many researchers with interest in particular domains. We believe that our database will inform and enable many detailed studies among other researchers, and that is the primary aim of our work.

Comment:

The experimental methods are described predominantly by referencing earlier work. It is difficult to be confident that these earlier papers provide a complete description of what was done, e.g. of the phage screening and ELISA protocols. This is particularly true because different papers

are cited in the text vs. method sections (e.g. Teyra 2017 vs. Huang & Sidhu 2011). Were experimental variables such as concentrations, buffers, and wash stringencies exactly as in these earlier works? It would be good to associate this large data set unambiguously with the protocol(s) that generated it.

Response:

We appreciate the reviewer's concern for detailing the methods as accurately as possible. Fortunately, in this case, the citation in the methods section (Huang & Sidhu 2011) is a detailed Methods paper, which we wrote precisely to establish and publish precise protocols for a standardized high-throughput pipeline for specificity profiling by peptide-phage display, and consequently, this reference very accurately details the phage screening and ELISA protocols used in our study. The second reference is a recent paper from our group where this high-throughput methodology was applied (Teyra, 2017). Since the experimental procedures are exactly the same, and the Huang & Sidhu Methods reports these in detail, we decided to keep the methods section short by referring to the Methods paper. We clarify this in the methods section as follows: "The exact protocol carried out in this study has been published previously (Huang & Sidhu, 2011)"

Comment:

The organization of the article could be improved. The text is redundant in places. Also, material suitable for the Introduction or Discussion appears in the main text, and aspects of the methods appear in the main text but not the methods section.

Response:

We have tried to reduce redundancy in the paper. Each subsection of the Results contains a brief conclusion at the end. In addition, some sections of the Results require a preliminary introduction necessary to understand the analysis and results, since this is a large-scale study involving many PRMs and comparison with large motif and structural data sets.

We agree that there are some aspects of the methods that appear in the Results section and we have rearranged some of these into the Methods section (i.e RHI). However, we believe that the definitions of the peptide library, PWM matrix, or SP score are necessary in the main text to better understand the results, given that this paper serves as an introduction to a very large database that will hopefully be used by many diverse members of the research community, including, for example, cell or structural biologists who study particular proteins but may not be

familiar with many of the methods and terms that are used in our large-scale studies. Consequently, we prefer to err on the side of caution and explain the work in a manner that a broad audience will be able to access, with the understanding that this may create some redundancy for those who are experts in large-scale profile studies of this type.

Comment:

Some of the supplementary materials have extra sheets in the excel file that don't appear intended for release. The authors should go through all supplementary tables and ensure that the terms used in the headings are precisely defined, with an equation if appropriate (all scores or ratios that are reported should be rigorously defined).

Response:

We ensured that the headings and data presentation in the supplementary are of equal quality and clarity as the main text and figures.

Comment:

The authors have the wrong year in this citation: Davey NE, Roey KV, Weatheritt RJ, Toedt G, Uyar B, Altenberg B, Budd A, Diella F, Dinkel H & Gibson TJ (2011) Attributes of short linear motifs. *Mol. Biosyst.* 8: 268-281

Response:

The error has been corrected and does not affect the other references. This particular article may have been stored in our reference manager library when it was accepted (2011) although it was published in 2012.

Reviewer #3:

Comments:

This manuscript reports a large scale survey of peptide ligands by phage display with many different protein domains, together with a derived database where users can explore the data. Now that it is clear that a huge amount of cell regulation involves short linear motifs (SLiMS), medium and high throughput methods are essential to screen for candidate motifs that can be subjected to low throughput validation. Phage display is an essential tool for this endeavour. The resource prm-db built upon the experimental work presented here will be explored by (increasing numbers of) experimentalists seeking for motif interactors in their systems. It might

also prove useful for annotated motif resources such as ELM to help in motif refinement, especially when only a couple of true instances are known. Overall, this is valuable work for the SLiM field.

One issue with phage display is that there can be a hydrophobic/Tryptophan bias in the recovered peptides. Importantly, this is addressed nicely in the discussion. There certainly are plenty of W-containing SLiMs such as DPW and WRPW and it is true that W is present more often than would be expected by chance. And many of the phage display motifs in the resource don't have W enrichment so then there is no issue. It is good, though, that researchers are properly informed.

I have no revisions to request.

Response:

We thank the reviewer for their comments and for acknowledging the quality of our work and its importance to the research community. With the revisions outlined above, we hope that we can now publish the manuscript and release the associated online database for use by the general research community. As noted by all three reviewers, the work will be of value to a broad range of researchers. As with any large-scale study, there are certainly many other analyses that can and should be done, and we look forward to seeing what use other researchers will make of our compiled database.

Thank you for sending us your revised manuscript. We have now heard back from the reviewer who was asked to evaluate your revised study. As you will see below, the reviewer is satisfied with the modifications made and is supportive of publication. As such, I am glad to inform you that we can soon accept your manuscript for publication, pending some minor editorial issues listed below.

REFEREE REPORTS

Reviewer #2:

The authors have satisfactorily addressed the points raised and also added new analyses that make the paper more interesting. This work is a valuable contribution as a research resource and I strongly support publication at this time.

2nd Authors' Response to Reviewers**3rd Nov 2020**

The Authors have made the requested editorial changes.

Accepted**4th Nov 2020**

Thank you again for sending us your revised manuscript. We are now satisfied with the modifications made and I am pleased to inform you that your paper has been accepted for publication.

Corresponding Author Name: Sachdev Sidhu

Manuscript Number: MSB-19-9310